# Key Global Actions for Mycotoxin Management in Wheat and Other Small Grains

**DOI:** 10.3390/toxins13100725

**Published:** 2021-10-14

**Authors:** John F. Leslie, Antonio Moretti, Ákos Mesterházy, Maarten Ameye, Kris Audenaert, Pawan K. Singh, Florence Richard-Forget, Sofía N. Chulze, Emerson M. Del Ponte, Alemayehu Chala, Paola Battilani, Antonio F. Logrieco

**Affiliations:** 1Throckmorton Plant Sciences Center, Department of Plant Pathology, 1712 Claflin Avenue, Kansas State University, Manhattan, KS 66506, USA; jfl@ksu.edu; 2Institute of the Science of Food Production, National Research Council (CNR-ISPA), Via Amendola 122/O, 70126 Bari, Italy; antonio.moretti@ispa.cnr.it; 3Cereal Research Non-Profit Ltd., Alsókikötő sor 9, H-6726 Szeged, Hungary; akos.mesterhazy@gabonakutato.hu; 4Department of Plant and Crops, Faculty of Bioscience Engineering, Ghent University, 9000 Ghent, Belgium; Maarten.Ameye@UGent.be (M.A.); Kris.Audenaert@UGent.be (K.A.); 5International Maize and Wheat Improvement Center (CIMMYT), Apdo. Postal 6-641, Mexico 06600, DF, Mexico; pk.singh@cgiar.org; 6INRAE, UR1264 Mycology and Food Safety (MycSA), F-33882 Villenave d’Ornon, France; florence.forget@inrae.fr; 7Research Institute on Mycology and Mycotoxicology (IMICO), National Scientific and Technical Research Council-National University of Río Cuarto (CONICET-UNRC), 5800 Río Cuarto, Córdoba, Argentina; schulze@exa.unrc.edu.ar; 8Departamento de Fitopatologia, Universidade Federal de Viçosa, Viçosa 36570-900, MG, Brazil; delponte@ufv.br; 9College of Agriculture, Hawassa University, P.O. Box 5, Hawassa 1000, Ethiopia; alemayehuchala@yahoo.com; 10Department of Sustainable Crop Production, Faculty of Agriculture, Food and Environmental Sciences, Universitá Cattolica del Sacro Cuore, via E. Parmense, 84-29122 Piacenza, Italy; paola.battilani@unicatt.it

**Keywords:** black point, deoxynivalenol, disease resistance, ergot, Fusarium Head Blight, nivalenol, Nominal Group discussion, post-harvest, trichothecenes, zearalenone

## Abstract

Mycotoxins in small grains are a significant and long-standing problem. These contaminants may be produced by members of several fungal genera, including *Alternaria*, *Aspergillus*, *Fusarium*, *Claviceps*, and *Penicillium*. Interventions that limit contamination can be made both pre-harvest and post-harvest. Many problems and strategies to control them and the toxins they produce are similar regardless of the location at which they are employed, while others are more common in some areas than in others. Increased knowledge of host-plant resistance, better agronomic methods, improved fungicide management, and better storage strategies all have application on a global basis. We summarize the major pre- and post-harvest control strategies currently in use. In the area of pre-harvest, these include resistant host lines, fungicides and their application guided by epidemiological models, and multiple cultural practices. In the area of post-harvest, drying, storage, cleaning and sorting, and some end-product processes were the most important at the global level. We also employed the Nominal Group discussion technique to identify and prioritize potential steps forward and to reduce problems associated with human and animal consumption of these grains. Identifying existing and potentially novel mechanisms to effectively manage mycotoxin problems in these grains is essential to ensure the safety of humans and domesticated animals that consume these grains.

## 1. Introduction

In 2019/2020, global small-grain production was: wheat—764 million metric tons (MMT), barley—156 MMT, oats—23 MMT, and rye—12 MMT [1]. High-quality grain is a critical component to global food security and meeting the challenge of feeding 9 billion people by 2050 [2]. Yet, as much as half of the grain harvested globally may be lost post-harvest to poor storage, waste and mycotoxin contamination, with a similar amount never harvested due to biotic and abiotic agents [3]. Mycotoxins produced on these grains can endanger the health of humans and domesticated animals [4,5,6,7]. The major toxins are well known [8], and their general distribution on a worldwide basis at least relatively well understood [9,10], although the distribution of these toxins could be changing in response to changes in climate and agricultural practices. The levels of these toxins in human foods and animal feeds often are limited by regulation within countries and in international trade. Reducing mycotoxin contamination has health and economic benefits and would provide more food, without more land being cultivated, for a hungry and ever-growing population.

Since the turn of the century, more than 10,000 papers (Web of Science) have been published on mycotoxins (deoxynivalenol, ergot, nivalenol, ochratoxin A, and zearalenone) and fungi that produce them on wheat and other small grains, yet the global levels of contamination have not been greatly reduced. Especially during more severe epidemics, potential control measures have limited impact and the reductions achieved may not suffice to decrease toxin contamination to less than regulatory limits. Thus, we think the 21st century’s record of managing toxin contamination needs to be looked at with a critical eye, and efforts to identify targets that could lead to significant reductions in mycotoxin contamination going forward.

Amongst diseases of small-grain cereals caused by toxigenic fungi, Fusarium Head Blight (FHB), also known as scab and caused by multiple species of *Fusarium*, is of particular concern due to the losses it causes and the toxins it can produce. *Fusarium* spp. can be easily found on any small-grain cereal growing anywhere in the world. Other non-*Fusarium* species, e.g., *Claviceps purpurea* (ergot) and *Alternaria* spp. (black point), can also lead to significant losses [11].

Scab epidemics have occurred in all major grain-growing regions in the world. In parts of China [12,13], nearly every year brings a new epidemic, and the FHB epidemic area has expanded to include the north and west winter wheat regions. Epidemics are also common in wheat-growing regions of South America [14,15,16,17,18]. In India and Pakistan, epidemics are occurring in the Himalayan foothills as maize production increases [19]. Reports from Africa are scarce, although regular epidemics occur in the irrigated wheat grown in the Orange River valley in South Africa [20]. In the United States, a new wave of epidemics began in 1993, and has continued at varying locations and with various degrees of severity ever since [21,22]. Canada has experienced similar problems in both Ontario [23] and in its western Great Plains provinces [24]. In Europe, epidemics, although variable, have become more frequent as *F. graminearum* has moved into regions in northern and central Europe previously dominated by *Fusarium culmorum* [25]. In Australia, an epidemic occurred in 2010 that was caused by both *F. graminearum* and *Fusarium pseudograminearum* [26].

Disease epidemics usually are accompanied by increased mycotoxin contamination, usually deoxynivalenol (DON), but sometimes zearalenone (ZEA) as well. The amount of toxin contamination can be considerable, but is not necessarily directly related to the severity of the disease outbreak. Disease resistance of wheat to different *Fusarium* species is governed by species-non-specific QTLs [27,28,29]. Most lines have no more than moderate resistance to FHB. Weather forecasts, if properly modeled, often provide accurate projections of disease severity, and sometimes of toxin production as well [30,31].

Measurements of toxin present are only as accurate as the chemical methods used to detect them. In some cases, mycotoxins, e.g., DON, are detoxified by plants by conjugating them to another molecule, without degrading them. These plant-conjugated toxins commonly are termed “masked mycotoxins” [32,33,34]. Risks posed by masked mycotoxins are unknown but are estimated based on the ease with which the conjugation reaction can be reversed and the conditions under which the reversal could occur. Rapid, effective simultaneous monitoring of multiple mycotoxins, e.g., [35], including masked mycotoxins, is an important goal for future research [36].

In this paper, we focus on toxins in small grains, primarily wheat, and the fungi, primarily *F. graminearum* (the major FHB pathogen), responsible for them. The initial sections are summaries of available control strategies, with some identification of regional differences. Differences between wheat and the other grains and problems caused by other fungi are incorporated into these sections, where information is available. Results from a Nominal Group discussion session held at the 2nd International MycoKey Conference in Wuhan, China (September 2018) are then presented to identify critical areas in which future efforts to control these diseases could be made.

## 2. Fungi

### 2.1. Fusarium

#### 2.1.1. Taxonomy and Geographic Distribution

*Fusarium* is a large, diverse fungal genus [37,38] that contains many strains capable of synthesizing mycotoxins [39,40]. The *Fusarium* species most commonly associated with FHB is *F. graminearum* [41], but other *Fusarium* species, including *F. acuminatum*, *F. avenaceum*, *F. culmorum*, *F. equiseti*, *F. incarnatum* (formerly *F. semitectum*), *F. langsethiae*, *F. oxysporum*, *F. poae*, *F. proliferatum*, *F. solani*, *F. sporotrichioides*, *F. subglutinans*, *F. tricinctum*, and *F. verticillioides*, have been widely isolated from infected wheat kernels [38,42,43,44]. Most of these species are neither common on small grains nor regarded as pathogenic towards them, although some may produce mycotoxins [45,46]. *Fusarium graminearum* is a hemibiotroph, as it is a biotroph in the early stages of infection and a necrotroph at later stages [47]. *Fusarium graminearum sensu lato* is a set of 16 phylogenetically distinguishable groups that are morphologically indistinguishable and often termed phylogenetic species [48,49,50,51,52]. *F. graminearum sensu stricto* occurs worldwide, is the group on which the most research has been conducted, and is the group to which the name *F. graminearum*, as used in this article, applies unless otherwise stated. It is widespread globally [53] and is usually the dominant species on wheat in North America [54,55] and Europe [56,57,58]. The species can be further subdivided into populations associated with the trichothecene produced—nivalenol (NIV), 3-acetyldeoxynivalenol (3-ADON), 15-acetyldeoxynivalenol (15-ADON), and NX-2 [50,59,60].

FHB epidemics in Latin America usually are caused by members of a 15-ADON-producing population of *F. graminearum* [61,62,63]. In southern Brazil, the 15-ADON population dominates (83% of the population), followed by the *F. meridionale* (13%), *F. asiaticum* (0.4%) and *F. cortaderiae* (2.5%) phylogenetic species, which primarily produce NIV. In Uruguay, the 15-ADON type again dominates (86% of the population), with strains of the *F. asiaticum, F. brasilicum, F. cortaderiae*, and *F. austroamericanum* phylogenetic species also detected [64]. In Argentina, the 15-ADON type again dominates [63], although 3-ADON-producing strain populations have also been detected [65].

Wheat in Asia is commonly contaminated with *F. graminearum*. In Iran, *F. graminearum* was the dominant species, with the 15-ADON type the most common in the western part of the country and a NIV-producing type the most common in the east [66]. In Northern India, *F. graminearum* was the most prevalent and the most pathogenic of the six *Fusarium* species identified [67]. No phylogenetic species were resolved and no toxin-related strain types were identified. In Korea, a NIV-producing population of the *F. asiaticum* phylogenetic species dominated on wheat in six provinces [68].

More work has been performed in China than in any other country in Asia. Five *Fusarium* species were identified in 15 provinces that included most of the wheat-growing region. *F. asiaticum* and *F. graminearum* dominated the southern and northern parts of the country, respectively, with *F. meridionale* found only in southwest region. All of the *F. graminearum* strains produced 15-ADON. In *F. asiaticum*, 3-ADON producers were common in the middle and lower parts of Yangtze River Valley, and NIV producers were the most common in the upper valley [13]. Similar results also were obtained in Japan, with NIV producers from *F. asiaticum* most common in the south and 3-ADON producers from *F. graminearum* most common in the north [69]. *Fusarium asiaticum* appears to dominate on wheat grown in a wheat–rice rotation, while *F. graminearum* dominates in wheat–maize rotations [70]. Differences in these cropping systems could explain the differences observed in the northern and southern parts of both China and Japan.

#### 2.1.2. Fusarium Head Blight Disease Process

At plant flowering, *Fusarium* ascospores are ejected from perithecia formed on plant debris [71], and spread through rain and wind to external anthers and outer glumes [72,73,74]. Although plant debris from previous crops is the most commonly considered source of inoculum, *F. graminearum* also can colonize numerous other gramineous and weed hosts that may provide an inoculum reservoir [75,76,77]. Fungal ascospore discharge requires water and light, with infection requiring a plant flower that is receptive to the spore [78,79]. Macroconidia of *F. graminearum* also can infect crops. In barley, the infection process is variety specific and somewhat different from that in wheat [80,81]. Infection timing can affect both disease severity and toxin accumulation [82]. In wheat, DON is an important virulence factor, and strains that produce DON are more virulent than those that do not [80,83,84]. Other virulence factors, e.g., the secretion of hydrolytic enzymes, also are associated with early stages of FHB infection [85].

The early stages of *Fusarium* infection are usually macroscopically symptomless [86]. The death of the ear axis results in bleached, shriveled grains without visual scabby symptoms. A few days after infection, invasive *Fusarium* mycelia spread throughout the spikelet, down into the rachial node and ultimately up and down the rachis as FHB symptoms become clear [87]. Infected spikelets first appear water soaked, then lose their chlorophyll and become straw colored. In more resistant wheat heads, the tracheas remain functional and the yield loss is smaller. In warm humid weather, pinkish-red mycelium and conidia develop in the infected spikelets. The infection spreads to adjacent spikelets and then throughout the entire head. Epidemics are enhanced by warm temperatures and high relative humidity (>70%), and/or frequent precipitation during heading and flowering [88], with the susceptibility window usually open for 8–9 days [89].

Fusarium Head Blight reduces wheat yield quantity and quality through the selective loss of albumin, gluten proteins and starch in the grain [90,91,92,93]. The infected grain also can be contaminated with mycotoxins, e.g., ZEA and the trichothecenes DON and NIV and their derivatives [21,94]. These toxins render the crop unsafe for consumption by humans and domesticated animals.

#### 2.1.3. Black Point

*Fusarium proliferatum* is associated with black point of wheat in North America [95,96], Argentina [97], China [98,99], Italy [100], and Nepal [101]. The resulting disease is phenotypically similar to that caused by other fungal agents, e.g., *Alternaria* spp., associated with the disease [96]. *F. proliferatum* is a well-known pathogen of numerous crops, most importantly maize [38]. Little work with isolates of *F. proliferatum* recovered from wheat has been performed beyond their identification as the causal agent of the disease, and it is not known if there are populations within the species that differ in their ability to cause this disease. Co-infection of *F. proliferatum* with *Fusarium* species that cause FHB has been documented [100].

### 2.2. Alternaria

*Alternaria* is a large complex genus [102] that is commonly recovered from wheat worldwide, some of whose members cause black point on grains. The taxonomy of the genus is not settled and the current entity might need to be split into multiple genera [103,104,105,106]. *Alternaria alternata* is the most commonly identified species, but its taxonomy can be complicated by the presence of pathotypes or *formae specialis*, which indicate plant pathogenicity [107,108,109], that provide little insight into phylogenetic relatedness or ability to produce toxins.

*Alternaria* species produce several mycotoxins such as tenuazonic acid, alternariol, alternariol-monomethyl ether, and altenuene, which have hematotoxic, genotoxic and mutagenic activities. Species-specific mycotoxin profiles have not been identified. However, *A. alternata*, *A. tenuissima* and *A. arborescens*, are all known to produce alternariol, alternariol-monomethyl ether, altertoxins I, II and III, and tenuazonic acid [110,111,112]. In contrast, strains of *A. infectoria* do not produce any of these toxins [111,113], although they can cause black point disease [114]. The *Alternaria* species commonly associated with black point disease on wheat, barley and oats are *A. alternata*, *A. tenuissima*, *A. arborescens*, *Alternaria mali* and *A. infectoria* [113,114,115,116]. On the grain, dark brown to black spots develop and can potentially cover the entire surface of the kernel. When severely infected grain is milled, undesirable black specks appear in the flour and flour derivatives [117]. Flour quality and nutritional value also are lower due to decreased starch and the loss of other important compounds [118].

### 2.3. Claviceps

The genus *Claviceps* contains approximately 60 species in four sections, with strains from all species capable of infecting ovaries of grasses and producing one or more ergot alkaloids. *Claviceps pupurea* is the primary cause of ergot disease and the contamination of rye and other small grains with ergot alkaloids [119]. Yet, in the United States alone, *C. purpurea* can colonize more than 160 different grass species [120]. Ergot of other crops, e.g., sorghum and pearl millet, have a different causal agent that does not contaminate grain with the same alkaloids [121,122,123,124,125,126].

#### Ergot Disease Process

*Claviceps purpurea* is common in geographic areas with cool, damp climatic conditions, [127,128]. In warmer climates, such as the southeastern U.S., sclerotia are colonized by other fungi and do not survive well [129]. Ergot usually is a bigger problem on open-pollinated grass crops, e.g., rye, than it is on self-pollinated crops such as wheat and barley. Ergot rarely occurs on oats. Infection occurs when a fungal spore lands on the (usually unfertilized) floret or ovary of a grass plant. Within five days, the floret begins to ooze a yellowish mucus, termed honeydew, which contains sugars and numerous conidial spores that can be dispersed by rain or by insects that touch the honeydew. Honeydew production continues throughout flowering. Honeydew on grain can gum up harvesting equipment and require extensive cleaning. As the ovary enlarges, honeydew production ends and a purplish-black sclerotium (or ergot body) replaces the plant’s seed. Ergot bodies are relatively large (up to 10× larger than the seed they are replacing) and emerge from the glumes. They have a hard exterior rind that is black to purple in color and contain alkaloids that may function as mycotoxins. Ergot bodies usually are physically separated by sorting from the rest of the grain based on size. A cold winter is required for ergot body germination, and moisture is needed in the spring for ergot body germination and the production of the first round of infecting spores. Cool, wet conditions usually lengthen the time during which flowering occurs and can also lengthen the time in the honeydew stage when additional contamination may occur. Ergot bodies can persist for a year in field soil, so rotating out of a susceptible crop often is the simplest control method.

### 2.4. Aspergillus

*Aspergillus* is a large fungal genus with at least several hundred species [130,131]. Of these, two, *Aspergillus ochraceous* and *Aspergillus flavus*, can be found in small grains, with *A. ochraceous* responsible for ochratoxin contamination and *A. flavus* for aflatoxin contamination. *A. ochraceous* may occur in the field; but for small grains, *A. flavus* is usually limited to poorly stored grain.

#### 2.4.1. *Aspergillus ochraceous*

*Aspergillus ochraceus* is a soilborne fungus that can infect a wide variety of crops including maize, peanuts, and small grains. The fungus is most common in temperate and tropical areas, and is not usually seen in more northerly regions, e.g., Canada and Scandinavia, that grow small grains. *Aspergillus westerdijkiae* and *Aspergillus steynii* are two additional, morphologically similar, species that have been split from *A. ochraceous* [132], and also can colonize small grains and produce ochratoxin A (OTA) [133]. Infections often begin in the field, but losses to plant disease caused by this fungus usually are minimal. Most losses attributable to *A. ochraceus* result from the production of OTA in storage.

#### 2.4.2. *Aspergillus flavus*

*Aspergillus flavus* is a globally widespread saprophyte and plant pathogen [134]. This fungus commonly is soilborne and is most abundant in temperate and tropical areas. It is most widely known as a pathogen of maize, peanuts and tree nuts and for its ability to produce the aflatoxin mycotoxins [135], but it can colonize almost any improperly stored grain, seed or foodstuff. *Aspergillus flavus* is usually not an important field contaminant of wheat [136,137,138]. It can be problematic in storage [136,139], however, with the rate of contamination potentially increasing with time [140,141].

### 2.5. Penicillium

Many species of *Penicillium* can be recovered from small grains, and distinguishing them can be difficult [142]. *Penicillium verrucosum* and *Penicillium nordicum* are the only species known to produce OTA, with *P. verrucosum*, but not *P. nordicum*, capable of colonizing small grains. *P. verrucosum* is the major producer of OTA in small grains in Canada, northern Europe and southern South America, and grain infestation > 7% with *P. verrucosum* is correlated with contamination with OTA [143]. This fungus is most common in cold climates and is almost always associated with small grains such as wheat, barley, oats and rye [133]. *P. verrucosum* causes most problems post-harvest when the grain is improperly stored, usually too wet (water content > 16%) [144,145], enabling OTA biosynthesis to occur.

## 3. Toxins

### 3.1. Trichothecenes

Trichothecenes are a large diverse class of toxins produced by multiple fungal genera, including both plant and animal pathogens [146]. The trichothecenes most commonly produced by *F. graminearum* are DON and NIV, although a new trichothecene, NX-2, has been reported to be produced by a novel subpopulation [147]. DON, also known as vomitoxin, is a type B trichothecene that can disrupt eukaryotic protein biosynthesis [39,148,149]. 3-ADON and 15-ADON are closely related chemically to DON and are usually included with DON in determining toxicity, even though the strains that produce them may belong to different natural populations or phylogenetic species [148,150,151,152,153,154,155]. Trichothecenes cause a wide range of acute and chronic toxic problems in humans and domesticated animals. These problems include food poisoning symptoms such as diarrhea, vomiting, abdominal pain and headaches in humans, and a broad array of problems in animals, including emaciation, vomiting, and reduced productivity or weight gain [39,148,149,156]. DON is an important factor in the virulence of *F. graminearum* on small grains [83,157], but NIV is not [158]. NIV has a role in virulence on maize [159] and is more toxic towards animals than is DON [160]. Plants sometimes glycosylate trichothecenes to reduce their in situ toxicity and thereby “mask” their presence in traditional chemical screens [32,161]. The glycosylation is readily reversed when the compound is consumed and its original toxicity is then restored as well [32,162].

DON is an intermediate in the trichothecene biosynthetic pathway that terminates with NIV [39], and strains usually produce one or the other of these two toxins, but not both. Genotype assays for toxin production are common but must be used with care as they may predict production of the wrong toxin, or even both toxins when neither is produced [163,164]. It is important to use the term “genotype” to describe the classes resulting from a genetic test and to reserve the term “chemotype” for use only when chemical data on the production of a toxin are available.

Geographically, DON, as either 3-ADON or 15-ADON, is the most widespread and can be found in small grains grown worldwide, e.g., Argentina [165], Brazil [44,166,167], China [168,169], India [6], and Italy [170]. Testing for NIV is much less common than is testing for DON, although it too has been reported from multiple countries including Argentina [171], Australia [172], Brazil [60,173], Canada [174], China [175], Germany [176], Iran [177], Italy [178], Japan [179], Korea [180], Poland [181], and the United Kingdom [182]. There are many more papers describing strains with NIV genotypes than there are those documenting NIV contamination of field material, which suggests that the geographic spread of NIV contamination may be wider than currently documented. Permitted levels of DON and other trichothecenes in various human foods and animal feeds are regulated in most countries that either produce or import small grains [148]. Regulatory limits vary by both location and the intended use of the product, and are usually much lower for material destined for human consumption than for material destined for consumption by domesticated animals.

### 3.2. Zearalenone

Zearalenone (ZEA) is a non-steroidal mycoestrogen, also known as RAL or F-2, found worldwide [183]. It is not toxic in that exposure does not result in death, but rather is considered toxigenic because it can induce estrogenic syndromes in swine, horses, and some lab animals. A chemical derivative of ZEA, zearanol, is marketed commercially as a growth promotor for cattle in feed lots [184]. ZEA is produced by a number of *Fusarium* species [38,39], of which *F. graminearum* is the most prominent. Genes responsible for the proteins that synthesize ZEA are found in a six-gene cluster and include two polyketide synthases [185,186]. Inactivation of ZEA biosynthesis did not reduce the pathogenicity of the strain carrying the inactivated gene towards barley [187]. Expression of genes in the cluster is altered when *F. graminearum* infects wheat [188].

ZEA and its in vivo metabolites, although not steroids, act similarly to 17β-estradiol and can bind to estrogen receptors [189]. They can inhibit the secretion and release of steroid hormones, cause hyper-estrogenism in mammals, most notably pigs, disrupt estrous cycles, and reduce male fertility. Maize and wheat are the grains commonly contaminated by ZEA. Contaminated grain may contain DON in addition to ZEA as both compounds can be synthesized by the same strain. ZEA also is common worldwide, e.g., Brazil [44,105,190], and China [191]. ZEA is widely regulated in international trade, with regulatory limits usually tied to the target species consuming the contaminated food/feed.

### 3.3. Black Point-Associated Toxins

Black point of wheat is associated with a variety of fungi including several species of *Alternaria* and *F. proliferatum*. Of these many fungi, only some *Alternaria* species and *F. proliferatum* are reported to produce mycotoxins of potential concern for humans and domesticated animals.

#### 3.3.1. Alternaria-Associated Toxins

*Alternaria* spp. synthesize numerous secondary metabolites that are potential mycotoxins [192]. Risks posed by the most prominent *Alternaria* toxins—alternariol, alternariol monomethyl ether, tenuazonic acid, iso-tenuazonic acid, altertoxins, tentotoxin, and altenuene—were evaluated by the EFSA CONTAM panel in 2011 [193] and again in 2016 [194]. Of these, only four—alternariol, alternariol monomethyl ether, tenuazonic acid, and tentotoxin—are common in grain. Tenuazonic acid may be of greatest concern as it is produced at high levels in in vitro wheat grain cultures (up to 8750 mg/kg) and at lower levels on in vitro rice grain cultures [195], while the opposite pattern was observed for the benzopyrenes and perylene derivatives. There are little data available on toxicity of these chemicals or their mode of action, with the most data available on alternariol [196]. Two of the toxins, alternariol and alternariol monomethyl ether, may have potential effects at dietary levels and warrant additional toxicity studies, while tenuazonic acid and tentoxin were not thought to be sufficiently toxic at dietary levels to be of concern. Alternariol and alternariol monomethyl ether are mutagenic in bacteria and mammalian cells [197]. The *Alternaria* toxins are almost never regulated due to their lack of documented toxicity, despite their widespread presence across the food chain.

#### 3.3.2. Fumonisins

Fumonisins are well known as contaminants of maize [38,39]. There are numerous fumonisin analogues, of which B_1_ is the most widely distributed and most commonly studied [198]. Fumonisins inhibit sphingolipid biosynthesis and have diverse effects in humans and numerous domesticated animals [39,199]. The most prominent health problems associated with fumonisins in humans are esophageal cancer and neural tube defects in newborns. The most prominent diseases caused by fumonisins in animals include leukoencephalomalacia in horses, pulmonary edema in swine, and liver and kidney failure in laboratory rats and mice. Fumonisins are most commonly produced by *F. proliferatum* and *F. verticillioides* [38]. *F. proliferatum* is known to cause black point disease on wheat in multiple countries (Section 2.1.2.). Whenever it does so, there is a possibility that it may produce fumonisins to contaminate the wheat kernels as well [200]. Wheat naturally contaminated with fumonisins is known [97,98,100], but the number of such reports is relatively few and the levels of contamination detected generally low. While fumonisins are important mycotoxins, there is little evidence that they occur commonly at any significant level in wheat.

### 3.4. Ergot Alkaloids

Problems with ergot and the associated ergot alkaloids are reported in some Assyrian cuneiform tablets with major problems documented during the Middle Ages when ergotism was named St. Anthony’s fire [201]. The regular occurrence of ergot alkaloids in rye, triticale, wheat and other minor cereals cultivated in Europe and elsewhere continues to this day, e.g., Orlando et al. [202] and Bryła et al. [203], with rye the most prominent crop for this problem.

Ergots synthesize numerous secondary metabolites, many of them alkaloids, which often are vasoconstrictors. In humans and domesticated animals they are associated with dry gangrene, especially of the extremities, hallucinations, tingling in the hands, fingers, feet and toes, burning or crawling sensations under the skin, miscarriage, and death [204,205]. The high bioactivity of these metabolites has made them quite useful to the biotechnology industry as a starting point for the synthesis of pharmaceuticals [126,205,206,207,208]. *Claviceps purpurea* produces three major groups of ergot alkaloids, clavine alkaloids, d-lysergic acid and its derivatives, and ergopeptines [208].

Fungal alkaloids in ergot sclerotia are distributed throughout the flour when the sclerotia are ground with the grain during processing, and these alkaloids are not destroyed by baking. When people eat alkaloid-contaminated bread or baked products, symptoms of ergotism may develop. Animals are affected by ergotism when they consume grain or feed containing sclerotia.

### 3.5. Ochratoxin A

OTA occurs widely on small grains and numerous other plants and plant residues. Its regulation as a health risk is becoming more widespread [209,210]. The toxin usually is synthesized by *A. ochreaceus* or *P. verrucosum*, with *A. ochraceous* more common in temperate and tropical climates and *P. verrucosum* more common in cooler regions. Ochratoxins are associated with acute renal failure, renal nephropathy, lesions, and acute tubular necrosis in humans [211,212]. They may also be human carcinogens (Group 2B) [213,214], and are known to be teratogenic, mutagenic, hepatotoxic and immunosuppressive [212,214]. In some areas, e.g., Algeria, nearly 70% of the wheat samples tested contained OTA [215]. Biosynthesis of OTA usually occurs in storage, so resistance to ochratoxin accumulation is not a common breeding target in wheat. Good storage practices, i.e., good-quality grain kept cool and dry, usually minimizes contamination with this toxin.

### 3.6. Aflatoxins

Aflatoxin B_1_ is the best known mycotoxin globally, and is most commonly associated with maize and peanuts. Aflatoxins are an uncommon contaminant in wheat, and are usually associated with wheat that has been stored improperly, e.g., Zahra et al. [216]. Problems also may occur under warm, humid field conditions if these conditions persist pre-harvest for an extended period of time [217]. Aflatoxins are associated with multiple medical abnormalities in humans and domesticated animals, most prominently with liver cancer and liver failure, but also with immune system depression and stunting in infants and children [218]. Since 2010, aflatoxins have been reported in wheat or wheat products from Algeria [136], Bangladesh [219], China [220], Iran [221], Israel [222], Kenya [223], Kuwait [224], Malaysia [225], Pakistan [226], Tunisia [227], and Turkey [228]. Such contamination most commonly occurs post-harvest rather than in the field.

## 4. Pre-Harvest

Contamination of crops with mycotoxins begins in the field. The earlier the problem is addressed in the growing season, the smaller it usually will be when the crop is harvested and stored. The pre-harvest integrated crop management package recommended to manage *Fusarium* mycotoxins in wheat [229,230] includes host resistance, fungicide and/or biocontrol application, planting and harvest times, and cultural practices, e.g., crop rotation, tillage, and fertilization. Collectively, these management practices have three main objectives: (i) reducing the amount of *Fusarium* inoculum in the field, (ii) preventing plant infection at flowering, and (iii) reducing disease spread within wheat ears. The resistance level of the crop being grown also affects all of the above cropping practices [229,231,232]. Most cultivars have no more than moderate resistance to *Fusarium*, so the combination of techniques used to reduce FHB depends heavily on available varieties and the conditions under which they are grown.

### 4.1. Wheat

#### 4.1.1. Fusarium Head Blight

##### Host Resistance to Disease and Toxin

Host resistance to FHB is quantitative and complicated, e.g., Yan et al. [233]. The current state of the field is summarized in recent reviews—one focusing on China [234], while for efforts elsewhere in the world Buerstmayr et al. (2009) [235], for research prior to 2009, and Buerstmayr et al. [27] for the most recent 10 years. Atanasoff [236] first recognized that there were different types of resistance to the disease, and five different types of resistance are now generally recognized: Type I—resistance to initial infection; Type II—resistance to spread within the plant; Type III—resistance to DON accumulation; Type IV—resistance to kernel infection; and Type V—disease tolerance [237,238]. Note that resistance to disease and resistance to toxin accumulation are not synonymous [17,239,240]! Practical identification of varieties with greater and lower levels of resistance to FHB followed shortly after the different types of resistance were recognized [241]. Resistance is quantitative in nature and QTLs for resistance have been localized to all wheat chromosomes [29]. Although a series of seven major QTLs (*fhb1–7*) have been identified and mapped, none of them provide complete resistance and their mode of action is not well understood even if the protein they encode is known. Most work has been performed with hexaploid bread wheat, with tetraploid durum wheat usually considered much less resistant than bread wheat [242].

Identifying and “stacking” multiple QTLs is the strategy pursued by most groups breeding for resistance worldwide. Resistance sometimes does not carry over from one location to another (or from one research group to another). This problem arises from the difficulty of getting regular repeatable challenges to material as it is being screened, and possibly from G×E interactions as well. With research groups active on all continents, breeding efforts are not necessarily well coordinated. CIMMYT (cimmyt.org) manages a large network of breeders and pathologists in diverse locations across the entire CGIAR system, and the U.S. Wheat and Barley Scab Initiative (scabusa.org) funds and coordinates much of the activity in the United States.

Resistance to DON may result from degradation of the toxin [243,244] or to glycosylation to form a non-toxic deoxynivalenol-3-glucoside (DON-3G) [245]. This glycolytic bond can be broken under acid conditions and free, toxic DON released. Since DON is a virulence factor, resistance to its accumulation improves overall resistance to FHB [246]. Some strains that cause FHB may produce toxins other than DON, e.g., NIV or ZEA, so host resistance to toxin accumulation should be checked for multiple toxins. Resistance is usually not limited to a particular species, but is instead global in nature with increased resistance to all *Fusarium* species known to cause the disease [247,248,249]. Increasing host resistance to FHB is likely to select for an overall increase in virulence of the *F. graminearum* population [250]. Multiple lines with moderate disease resistance are widely available and when combined with an effective fungicide treatment (see “Fungicides” section below) usually provide adequate disease control except in years with severe disease epidemics.

FHB results from a series of complicated interactions [251]. A more resistant cultivar is more easily protected with fungicides, can better withstand higher levels of inoculum from a previous crop, and better tolerate suboptimal weather and agronomic conditions associated with fertilizer application and/or tillage. Thus, the resistance level is critical for the impact of different agents that determine if FHB occurs and its severity [252,253].

##### Fusarium Head Blight and Deoxynivalenol Forecasting Models

Multiple forecasting models are available for both FHB and DON accumulation [30], and in a few cases have been incorporated into decision support systems, e.g., Rossi et al. [254]. FAO publishes a forward-looking quarterly report, *Crop Prospects and Food Situation*, with cereals as the main focus [255]. Validated models that can accommodate weather fluctuations due to climate change are particularly important. All of the models incorporate meteorological data, e.g., relative humidity, rain and temperature, and some include plant development stages. Host resistance usually is not a modeling parameter, e.g., Giroux et al. [256], Lecerf et al. [257], and Liu et al. [258], as the necessary data generally are not available, even though susceptibility to disease can influence forecast results [259]. Current models assume uniform host susceptibility and are successful at a rate of 70–80% or higher. Forecast failures usually are attributed to higher resistance in the variety cultivated or to extraordinary ecological conditions. With current models, increased host disease resistance decreases the reliability of the forecast.

The DONcast^®^ model was first developed for use with wheat grown in Ontario, Canada [260] and estimates DON contamination rather than FHB disease incidence. This model predicts DON levels at harvest based on rainfall and temperature around flowering, cultivar planted and the previous crop in the field. The model is simple to use and can be set to send an SMS alert to farmers when conditions are right for FHB to occur. The DONcast^®^ model is delivered through Weather Innovations [261] in Canada. In Europe, DONcast^®^ is available through Bayer CropScience.

Both empirical and mechanistic forecasting models for disease incidence and DON accumulation have been developed for use in Europe [262,263,264,265]. Some models are specific for particular geographic areas while others have much broader applicability. In cross-validation studies [31], several models yielded similar, but complementary, results, which suggests that using multiple models could strengthen conclusions obtained from any single model.

In South America, the first model for FHB was for wheat in the Pampas Region of Argentina [266], where it has been successfully deployed since the 2005–2006 wheat-growing season [16]. In Uruguay, the DONcast^®^ model has been validated and adapted to the local conditions. In Brazil, the Sisalert model was developed in the early 2000s and predicts the daily risk of infection based on the rain and temperature variables beginning at flowering [267]. In the United States, results from an updated version of the model first proposed by De Wolf et al. [88] are available [268] for all wheat-growing regions east of the Rocky Mountains. This model relies on weather during the seven days prior to flowering and can be used by farmers to determine whether to apply fungicide to control the disease. The model estimates the risk that FHB severity exceeds 10% and is approximately 75% accurate.

##### Cultural Practices

Many studies of cultural practices focus on a single factor, often using extreme conditions that do not necessarily reflect conditions found in the field. For example, fungicide tests normally are made with a susceptible cultivar to identify anti-fungals that can protect the most sensitive material cultivated. In the context of crop rotation, host resistance is a critical factor but varieties with different resistance levels seldom are included. Studies of tillage, and planting and harvest dates suffer from similar problems. Highly susceptible cultivars cannot be protected successfully even when the best fungicides and management strategies are employed [252,253]. Moderately resistant cultivars are the best that are currently available, and they can be seriously impacted in a major epidemic. The resistance of the available cultivars is most effective when they are part of a well-organized integrated crop management system that also considers crop rotation, tillage, residue management, soil fertility, and irrigation, and uses a disease forecasting model(s) to help manage fungicide and pesticide applications.

##### Crop Rotation

Crop rotation plays an important role in determining disease level in the crop. The function of crop rotation is to reduce the amount of primary inoculum available during the cropping season [269,270]. For FHB, the most important crop rotation question has been whether maize was grown in the field in the previous year [271,272,273,274], and the answer can affect both disease incidence and the amount of DON produced. Fungal perithecia are produced on plant residue left in the field from the previous cropping season. Differences in susceptibility to *F. graminearum* of the plant residue could affect the number of perithecia formed and the speed at which they form. The amount of moisture present during the off-season can determine the number of perithecia and their ability to produce and eject spores, which rely on water pressure to eject spores from the perithecia [275]. Similar problems, i.e., inoculum material that is too dry, can occur even in research nurseries that use maize stalk residues as a source of inoculum, e.g., Mesterházy et al. [276]. This problem is common to all regions where wheat and maize are grown either concurrently (in nearby fields) or in rotation with one another.

Perithecia of *F. graminearum* can be recovered from many crops [38], but these crops are not all equally good as hosts of the fungus. The risk of DON contamination increased 15-fold when wheat followed maize in the field [277], but this degree of increase is not universally observed [18]. Levels following canola, sorghum or soybean were no more than a quarter of the levels observed when following maize. Most recommendations are for a non-host crop as part of a rotation to separate two cereal crops, e.g., Dill-Macky and Jones [271] and Krebs et al. [278]. Soybean, which is a poor host for *F. graminearum* [279], is a commonly recommended choice in South America and Asia. Sugar beet was effective as a break crop in Poland [230]. Rotation may have little impact, however, if environmental conditions favor an epidemic, e.g., Kukedi [280].

In the subtropics of Brazil, crop rotation is of lower importance. The year-round presence of regional stubble- and grass-borne fungal growth provides an effectively unlimited source of airborne inoculum for epidemics. This more-or-less continuous supply of inoculum overshadows the effect of crop rotation in disease control [14], and different planting dates rather than crop rotation are used to reduce risks. Tillage (see “Tillage” section below) also can help determine the impact of rotation on disease pressure in succeeding crops, and decisions on tillage and crop rotation usually need to be made concurrently. In conclusion, the cropping history of a field is an important factor, but the amount of available fungal biomass, weather, and variety resistance can amplify or diminish the impact of the cropping history.

##### Tillage

As with crop rotation, tillage affects the amount of primary inoculum present. The effectiveness of tillage often is intertwined with the previous crop planted and the disease resistance of the currently planted variety. There are multiple reports that plowing can have as much impact on mycotoxin contamination as planting a resistant variety [281,282,283]. The effects are additive and it is possible to combine resistance with crop rotation and tillage practices to greatly reduce toxin contamination [253].

Minimum tillage often is preferred because it reduces run-off, retains organic matter in the soil, is less expensive, and is more efficient in terms of time and equipment management. Minimum tillage, however, increases the potential for large amounts of primary inoculum in the succeeding year if the previous crop was one that could be colonized by *F. graminearum*, e.g., a small grain or maize [281,284]. Infected crop residue that remains on the soil surface in the minimum tillage scenario is ideal for fungal growth and spore production [285].

Plowing buries infected residue, reduces the formation of fungal perithecia and ascospores that serve as the primary disease inoculum, and is as effective in oats as it is in wheat [286]. Champeil et al. [287] and Imathiu et al. [288] both argue for plowing as it destroys any infected residue from previous maize or small-grain crops. Jacobsen [289] concluded that tillage was effective only if a minimal amount of infected debris remained on the soil surface. The Asian experience is similar to that in Europe and the Americas, with rotary tillage and plowing effectively reducing primary inoculum. Deep ploughing can reduce disease incidence by >50%, but the expense of the process limits its application

##### Planting Date

Planting date often determines flowering date, and the environmental conditions at flowering are critical for the occurrence of FHB. The goal of this strategy is for flowering to occur when weather conditions are less favorable for disease onset. Unfortunately, climatic conditions at flowering cannot be predicted, so the efficacy of this strategy varies. In temperate regions, there are reports that early sowing dates are preferable [71,290] while others detected no significant effect [291]. In some cases, farmers will stagger planting dates with the goal of having at least some of the crop flower when the weather is not favorable for disease onset. In the subtropics of Brazil, there is a continuous and effectively unlimited source of airborne inoculum of *F. graminearum*, and planting dates are spread out to reduce disease risks [14].

##### Nitrogen Fertilization

The amount of nitrogen applied can affect the health of plants, with healthier plants often less susceptible to both disease and toxin contamination. Neither Krnjaja et al. [292] nor Yoshida et al. [293] found a significant effect between medium and higher nitrogen application rate on DON contamination levels. With artificial inoculation at higher levels of nitrogen DON contamination increased, but ZEA levels were not meaningfully altered. Thus, high levels of nitrogen fertilization could be risky under epidemic conditions when significant amounts of natural infection occur in susceptible cultivars. Lemmens et al. [294] tested nitrogen application rates between 0 and 160 kg/ha. They found a significant increase in DON contamination as the amount of nitrogen applied increased from 0 to 80 kg/ha, but no further increase at nitrogen application levels above 80 kg/ha. This result is consistent with those of Krnjaja et al. [292], since in those tests neither resistant nor moderately resistant cultivars were included. In general, fertilization does not have a primary role in determining the amount of toxin produced. Instead, tillage and plant debris have much more important roles.

##### Irrigation

Generally, in FHB-endemic regions, wheat is not normally grown with irrigation, although the Orange River valley in South Africa is an important and notable exception [20]. Irrigation can be as important as rain in providing conditions conducive for FHB [289]. When conditions are dry, the crop should be irrigated approximately one week before flowering and again approximately 10 days after mid-flowering to reduce the disease risk. The negative impact of increased disease and toxin contamination that can accompany irrigation is more important for susceptible host lines than it is for those with some disease resistance [295].

##### Fungicides

Fungicides are a critical and commonly used tool for managing FHB [296]. The fungicide, the application timing, method and rate, and the host line being protected all play an important role in determining the degree of disease and/or toxin reduction that occurs and thus the efficacy of the fungicide applied. As multiple populations and species of *Fusarium* can cause FHB, the differential fungicide sensitivity of these groups can also be an important determinant of fungicide efficacy [297,298,299]. Fungicides to reduce FHB are usually applied within a few (commonly four) days of anthesis [300,301]. Disease prediction models (see section on FHB and DON modelling above) often are used to determine the risk of fungal infection and whether the expense of a fungicide application is warranted to protect the crop [302]. In Africa, the cost and timely availability of fungicides often limits their use, especially by small-holder farmers.

Fungicides commonly used to control FHB, not all are equally effective [303,304,305], include carbendazim, triazoles, e.g., metconazole, tebuconazole and prothioconazole, and strobilurins, e.g., azoxystrobin, or combinations of two or more of these compounds [21,306]. In particular, azoxystrobin alone should be avoided as its use can increase the amount of DON produced [305,307]. In China, carbendazim has been the primary fungicide since the 1970s, and >60% of the fungal population in Jiangsu Province was carbendazim resistant in 2016 [307]. Tebuconazole is now the primary fungicide with DON levels lowered by 38–79%, and no resistant isolates recovered from the field as of 2014 [308].

Application timing [297] and technology, with spray nozzles and application angles critical variables [253,309], are also very important in determining fungicide effectiveness. Limitations in application technology [21] also limit efficacy. Effective translocation of tebuconazole and prothioconazole from flag leaves to heads does not usually occur. Even within the head, translocation from a treated floret is approximately 15% to the next floret up in the head, while translocation one floret down or to a floret on the other side of the head is only 1–3%. Therefore, if at all possible, the entire head should be sprayed, because directly treated florets are the ones most likely to be protected [310,311].

Fungicides are more effective on resistant lines than they are on susceptible ones [231,252,253,309]. For example, an excellent fungicide applied to a highly susceptible cultivar could reduce the DON level by 70%, but this level could still exceed regulatory guidelines. Substitute a moderately resistant variety and toxin reduction is 95–98%, and the final toxin level is far below the regulatory limit. Thus, the same fungicide can have high, medium or low efficacy on cultivars with differing resistance. In general, the resistance level is a more powerful regulator of FHB than any other single measure [312].

##### Biological Control

Biological control is an attractive strategy for controlling plant diseases due to environmental and legislative limitations, e.g., Directive 2009/128/EC [313], on the use of chemical pesticides for this purpose [314]. Biocontrol agents usually are microorganisms whose activity may be in one or more of several broad areas, including antibiosis, competition, mycoparasitism, and indirect activation or enhancement of plant defense [315]. Biocontrol agents usually are used in pre-harvest applications, although post-harvest uses are also possible [316]. Numerous potential biological control agents have been tested in the lab, with the number that have succeeded in field trials smaller, the number of commercially available agents even smaller, and no biocontrol agents currently in widespread commercial use anywhere in the world [317,318,319]. Strains of numerous prokaryotes, including *Bacillus* [320,321,322], *Pseudomonas* [323,324], and *Streptomyces* [320,325] can reduce disease incidence and/or DON accumulation under field conditions. Effective fungi include *Clonostachys* [326,327], *Cryptococcus* [321], *Pythium* [328], and *Trichoderma* [328]. In at least some cases, the mode of action also has been studied [322,323].

To be commercially useful and successful, biocontrol agents face two significant hurdles. First, many of these organisms are most effective when applied as part of a mixture of different strains. Current regulations require each member of such consortia to be tested and registered separately, which can become quite expensive. Second, many biocontrol agents reduce disease or toxin levels by a significant amount, but perhaps not as much as required to satisfy regulatory mandate. For example, an agent that reduces toxin level by 80%, but leaves a toxin residue that is several fold above the regulatory limit is a clear scientific success, but not a commercial one unless combined with a complementary treatment to reduce the amount of toxin present to less than the regulatory threshold. A commercial product with these properties might be useful when the host plant already has moderate to high resistance or when the disease epidemic pressure is light. Adding another layer of complexity to effective biocontrol is the residential microbiome. For example, as listed in [329], two species of *Actinobacteria* are effective biological control agents for *F. graminearum*, but if *F. poae* is present, the biological control does not occur.

##### Insect Control

Grains damaged by insects are more prone to infection by toxigenic fungi as the wounds created by insect attack facilitate fungal entry. Insects can vector fungal spores and/or create conditions that favor fungal growth and mycotoxin production in the grain [330]. Therefore, effective insect pest management, through insecticides or otherwise, is essential to prevent the production/accumulation of mycotoxins in the grain [331].

##### Organic Farming

Edwards [182] did not find significant differences in DON levels between wheat samples from organic and conventional farms. Under low or moderate disease pressure, this result is expected; but with high disease pressure, the organic samples may have more DON.

#### 4.1.2. Control of Black Point

In breeding programs, black point-infected lines often occur. While these lines normally are discarded, there is variation in commercial lines for sensitivity to black point [332]. Black point is more severe in durum wheat than in hexaploid wheats. There is no specific fungal control for black point, but fungicides used to control FHB or leaf diseases also may reduce infection by *Alternaria*. More research is needed in this respect. Delays in harvest can allow more fungal growth and kernel discoloration. Modern grain sorting machines, e.g., Sortex, Buehler, Switzerland [333], separate black point-infected grains from sound grain and can lower the apparent fungal and mycotoxin levels in the final milled product. If the grain loss exceeds 20%, then the process is not economically viable.

### 4.2. Rye (Secale cereale)

Rye is a fast-growing cereal that is well adapted to poor sandy soils and grown in Europe, Asia, and North America as a spring or winter crop. Rye is the most important crop for ergot contamination. In rye harvested by combine, most of the ergot sclerotia remain with the grain, which necessitates post-harvest processing to remove the sclerotia. Increased planting of triticale on the better rye soils has reduced ergot problems somewhat. Rye plants also can be colonized by toxigenic *Fusarium* spp., with symptoms and toxin contamination similar to what is seen on wheat.

#### Control of Ergot

Openly pollinating cereals, e.g., rye, triticale, and hybrid wheat, are most exposed to ergot infection, while self-pollinating cultivars and species are largely protected. Ergot of rye causes yield reductions, and grading standards can reduce the value of harvested grain depending on the number of sclerotia present in the final product. Cultivars of wheat [334] and rye [204,335] differ significantly in their susceptibility to the disease, but immunity has not been identified. In hybrid rye, the susceptibility was more than three times higher than in open-pollinated cultivars, so there are opportunities to reduce susceptibility through resistance breeding, even though little progress has been achieved in breeding for host-plant resistance during the last decade [334].

Employing a cultivar with low susceptibility to the disease is a promising strategy to mitigate ergot alkaloid contamination, but is not the only available option. Crop rotation to a non-host crop, controlling grassy weeds that can serve as alternate hosts, tillage (to bury sclerotia), and selective harvesting of grain in heavily contaminated fields can all reduce the amount of ergot present in the grain [202,204]. Ergot bodies remain viable for up to a year on the soil surface, so rotating to a crop that is not susceptible to ergot contamination and/or applying fungicides to soil can reduce sclerotial germination and sporulation the following year [336]. Thus, multiple tactics are available to limit plant infection by ergot fungi and reduce field level contamination of grain with ergot alkaloids.

## 5. Post-Harvest

Post-harvest management methods are very similar for all small grains, thus we treat the general topic rather than breaking the analysis down by crop. Contamination may begin in the field and then be amplified during storage and processing if the grain is not handled properly [337]. A recent in-depth review [338] of this area with more detail and many more references is available. Our discussion is to orient the reader to some of the critical points.

### 5.1. Harvest

Harvest is a critical time, as there are many opportunities for grain damage and fungal infection as the grain transits from the field to storage [339]. Since toxin contamination is only rarely uniform between or even within fields, contamination should be checked in every truck load as it comes from the field. Combines should be sterilized and clean, cleaned again between fields, and set to minimize the number of broken kernels generated through the harvesting process. If precision harvesting, where one portion of the field is harvested separately from another, is to be employed, then the pattern to be followed should be identified in advance. Harvest preferably occurs when the grain has dried sufficiently to minimize or eliminate the need to dry it immediately after harvest. If the grain is wetter than can be safely stored, then it should go to a drying facility prior to storage to minimize grain deterioration.

Timely harvest can have an important impact on mycotoxin contamination in harvested grain. Delaying wheat harvest by a month, for example, increases the mean DON and ZEA levels by 10–25 fold [281]. Late harvests enable strains of storage pathogens, e.g., *Penicillium*, *Alternaria*, and *Aspergillus*, to infest the crop and establish themselves increasing the risk of toxin biosynthesis, especially if storage conditions are not optimal [289]. The only reason to recommend a delayed harvest is if the field is heavily contaminated with ergot. In such settings delaying the harvest may allow some of the ergot bodies to separate from the grain heads and fall to the ground rather than to be harvested along with the grain [334]. The feasibility of early harvests, which can again reduce fungal contamination, depends on the cost and availability of facilities to dry the grain to levels where it can be safely stored.

### 5.2. Grain Cleaning and Sorting

Grain cleaning may occur before the grain is placed in storage and then again just prior to milling or processing. The goal of pre-storage cleaning is generally to remove foreign material, e.g., dirt, stones and weed seeds, and inferior grain that is small, broken, shriveled or of low density. Grains infected with *Fusarium* are generally shriveled, relatively small, and weigh less than healthy grains, so infected and non-infected grains can be separated based on their physical characters. If the grain is infected later, however, the infected kernels are close to normal in size and test weight and the surface cell layers commonly have higher toxin levels than those found inside the grain. These grains are not as easy to separate from the healthy ones. In early tests, when wheat grains < 2 mm in diameter were removed, the DON concentration decreased by 83% (from an initial concentration of ~5 mg/kg), but 55% of the grain mass was lost [340]. More modest losses of grain mass (2–7%) resulting from removal of grains < 2.4 mm in diameter resulted in a reduction in DON levels by 22–27% [341]. Thus, the efficacy of cleaning protocols clearly varies by batch. Several mechanisms can be used including sieving, and gravity separation. Cleaning is not considered first-stage processing and the regulated toxin levels in cleaned grain are the same as those for unprocessed grain.

The material separated from the grain during cleaning is usually termed screenings. This material is of low value and the grain pieces in the screenings may be more contaminated with toxins. Prior to milling, a second round of more intense cleaning usually occurs. The grain may be washed with water, if it will be wet milled, or scoured, if it will be dry milled. These processes remove materials attached to the grain surface and also may remove parts of the outer layers of the grain. Current cleaning processes remove 7–50% of the toxin contaminating the grain [338,342] and are equally effective for trichothecenes, ZEA and OTA. The toxins removed are concentrated in the scourings that are the by-products of the cleaning process.

Optical scanning can sort out *Fusarium*-damaged grain and identify > 92% of the infected kernels based on the size, shape and color of the grains [343]. A round of optical sorting can remove ~10% of the grain, but the sorting equipment is relatively expensive to purchase and use. Manual removal of visibly damaged grain can remove as little as 1–2% of the total grain and lower toxin levels by 50–90% [344,345]. Optical scanning is not highly effective for removing OTA contaminated grain since this mycotoxin commonly is synthesized during storage post-harvest and infected grains may have no signs of abnormal development. Extensive scouring of the grain may be required to reduce OTA levels to less than regulatory thresholds [338,346,347].

#### Ergot

Sclerotia of *C. purpurea* are easily distinguished from the grain by both size and pigmentation. Modern grain cleaning and milling methods exclude sclerotia from material to be ground into flour [204,348]. It may not be possible to sort all of the contaminated material from heavily contaminated grain lots, in which case the grain should be buried [334,347]. Contaminated flour can still be found in rural areas of developing countries, although sclerotia can be readily separated from the grain based on visual observation and hand sorting, if necessary.

### 5.3. Drying, Storage and Decontamination

Drying grain to 10–14% moisture content, i.e., an *a_w_* < 0.70, is an acceptable norm for grain to be stored. Dried grain can be stored for six months to a year, depending on external factors such as temperature and humidity, which must suffice to prevent the growth of fungi and most insects. The drying process often is not uniform, with damp spots in the stored grain resulting from grain respiration leading to insect and fungal growth and damage to the grain [349,350]. As additional respiration occurs, the affected area enlarges and the cycle repeats itself. The biggest storage problems result from the grain being too wet, or becoming too wet during storage. Thus, accurate monitoring of moisture levels and other signs of respiration, e.g., CO_2_ levels, is critical to maintaining healthy grain in storage. If mycotoxin-producing fungi cannot grow then additional toxins will not be produced, although toxins already present usually will not be degraded either. Over-drying can damage the grain and make it easier for fungi to grow and produce toxin when the conditions are right. Overly dry grain also can result in lower-density grain and reduce its value since the grain usually is sold by weight.

The most important thing about grain storage is that mycotoxin levels will not increase if the grain is properly stored, but may increase if it is not. Thus, the lower the level of toxins in the grain when it is stored, the lower the levels in the grain when it is taken out of storage. There are numerous Extension Bulletins, and guides to best storage conditions available on the internet, that can be consulted for more detail, e.g., Grains Research and Development Corporation [351], Harner et al. [352], Payne [353], and Sadaka [354]. We summarize a few of the main points:
Dry grain to 10–14% moisture content as quickly as possible. In many cases, grain at harvest will already be this dry. Dry as much as is needed for safety, but do not dry any further as excessive drying reduces the monetary value of the grain, which is sold by weight. For example, grain with 10% moisture is worth approximately 4% less than grain with 13% moisture, although drier grain can be stored for a longer time than wetter grain.Store in a suitable bin. The bin should be aerated if storage is to be for more than six months, and also may need to be cooled if the grain is stored through the summer. Bins should: (i) hold the grain without leaks or spills, (ii) prevent rain, snow or soil moisture from reaching the grain, (iii) protect grain from rodents, birds, other pests, damage from fire and wind, and theft, (iv) permit effective insect control, and (v) have sufficient headspace above the stored grain for sampling, inspecting, ventilating and treating the grain.Control insects in as many ways as necessary. Potential controls include (i) interior bin and bin perimeter sprays, (ii) insecticides applied to the grain as it is transferred to the bin, (iii) special treatments to protect grain that is most exposed at the top of the bin, and (iv) fumigation to reduce ongoing infestations.Control temperature. The lower the temperature the more difficult it is for significant insect or fungal metabolism to occur. A winter temperature of ~5 °C is viewed as optimal. A sensor network in the bin can help detect respiratory hotspots, which may be problematic if left untreated. Grain cooling usually is accomplished by aeration with outside air that is at least 5 °C cooler than the grain itself.Aerate the grain. Aerating the grain reduces the opportunity for moisture to collect and helps keep the temperature uniform. Aerate, usually from the bottom of the bin, long enough so that if a moisture front forms it can move completely through the grain and out the top of the bin. The relative humidity of the air used for aeration varies with temperature, but usually should be <60% to prevent rehydration of already dried grain.

Numerous techniques to decontaminate grain have been tested with varying degrees of success. We discuss three—ozonation, steam and ammoniation—but others, including irradiation, cold plasma, steeping with organic acids, chlorine or sulfites, and heat, also have been tested, with more details available elsewhere [338,355,356,357,358,359].

#### 5.3.1. Ozonation

Ozone (O_3_) is a strong oxidant that can be applied as a gas or dissolved in water. It rapidly decomposes to O_2_, which leaves no harmful residue, and is classified as Generally Recognized As Safe (GRAS) by the Food and Drug Administration in the United States [360]. Recent reviews summarize much of what is known about the effectiveness of O_3_ in managing contaminating microorganisms and mycotoxins in wheat and other small grains [361,362,363].

Ozone can both degrade mycotoxins and kill the fungi that synthesize them. It oxidizes amino and sulfhydryl groups in proteins and fatty acids common in cell walls. The mechanisms by which O_3_ degrades mycotoxins are generally not well understood, although DON is degraded to less toxic compounds under laboratory conditions [364]. Mycotoxin degradation in naturally contaminated material may be less than 100% [365,366], with ZEA more susceptible to break down by O_3_ than DON [367]. Fungal growth, germination and sporulation of *Fusarium, Aspergillus* and *Penicillium* can all be limited or completely inhibited by ozone, thus preventing additional toxin biosynthesis after treatment [368,369]. In some cases, products made with grain that has been treated with ozone are better in quality than similar products made from untreated grain, e.g., Li et al. [364] and Zhu [370].

The efficacy of O_3_ treatment depends on a myriad of factors including, but not limited to: O_3_ concentration, exposure time, substrate, moisture content, pH, mode of application (gaseous or aqueous), and the fungal species present and their growth stage(s). The level of O_3_ applied and its format depends on the material being treated, with O_3_ more effective if the contamination is on or near the surface of the grain than if it is internal within the grain. A critical consideration is whether the treated grain is expected to be viable and germinate following treatment, e.g., for malting [368,371]. If so, then the time of exposure and the O_3_ concentration used are usually much less than if the grain is to be milled into flour or used for other products.

#### 5.3.2. Steam

DON can be degraded under some conditions at temperatures ≥ 100 °C. Steam can be used to reach these temperatures, but it is not usually used until after storage is complete since it hydrates the grain. Cooking wheat grain for breakfast cereal at 100 °C for 30–40 min does not reduce the amount of DON present unless the aqueous effluent is discarded. When it is discarded, then DON levels are reduced by >50% [372,373]. If bran is included in the breakfast cereal, then the toxin loss is 10–20% of that in the grain. Superheated steam (160–265 °C) can reduce DON by 25–75% in whole grain and by 25–60% in flour depending on temperature and the length of time of exposure to the steam [374,375]. Temperatures < 160° do not significantly reduce toxin levels [375]. Steam-treated flour is of lower quality due to protein (gluten) denaturation, increased water absorption by the dough, discoloration (browning), and gelatinization of some of the starch. This flour can still be used for some biscuits, cakes and pastries.

#### 5.3.3. Ammoniation

Ammoniation has been used to reduce contamination in maize, primarily to reduce the levels of aflatoxin and OTA [358]. In wheat, ammoniation can potentially degrade all of the toxin present in the grain [376] and up to 75% of the DON [376]. Factors such as temperature, moisture, pressure, duration and substrate all influence the efficacy of the process in both maize and wheat [377,378]. Other than for aflatoxins [379], the breakdown products of mycotoxins in ammoniated grain have not been well studied. The limited studies of DON degradation in wheat suggest that the relatively poorly described degradation products resulting from ammoniation should be less toxic than the original toxin [377].

### 5.4. Milling

Milling reduces grain to usable particles and segregates them into multiple fractions with different properties and values. Mills often perform a final, intense cleaning step before actually milling the grain and may debran the grain at this time as well. Partial debranning of 9–10% of the grain weight can reduce toxin content by 60%, with total debranning and a loss of up to 25% of the grain weight often not resulting in much additional (≤8%) toxin loss [380]. While milling does not change the amount of mycotoxins present, their redistribution can be quite significant [381]. Mycotoxins are more commonly found in the outer layers of the grain and fractions enriched for these layers, e.g., the bran, also are enriched for toxins [382]. There are two general types of milling—dry milling and wet milling. Dry milling usually results in flour and semolina products, while wet milling commonly is used to produce starch.

#### 5.4.1. Dry Milling

Most wheat is initially dry milled, with the grain separated into germ, bran, white flour, and semolina. Flour produced during the production of semolina is of lower quality than if flour is the end target of the milling process and also may be relatively more highly contaminated with mycotoxins. Bran-containing by-products of dry milling usually have higher levels of toxins than do white flour and semolina, which are derived from the endosperm. Fungal colonization patterns of the grain may affect where toxins are found and how they are distributed amongst the milled products [381], as can weather [383] and the cultivar grown [342,345,384]. Relative reduction in toxin content in the milled product does not appear to be affected by the initial toxin content of the grain [345,381]. In general, toxin levels in white flour are 50–80% of the toxin level found in the unmilled grain, and semolina toxin levels are ~60% of the values in the unmilled grain [338]. Results for wheat germ vary and are inconsistent, so monitoring germ for toxin levels post-milling is essential. OTA and aflatoxin appear to be reduced by similar or greater percentages [347,385,386], probably due to the preponderant occurrence of these toxins on the outer layers of the grain.

#### 5.4.2. Wet Milling

Wet milling is for the production of starch and results in gluten, bran, fiber and germ by-products. Wet milling may start with whole grain, but more commonly begins with white flour produced by dry milling. The steeping and washing steps in the wet milling process offer multiple opportunities to leach mycotoxins into the washing water, with levels of more water-soluble toxins reduced more dramatically than levels of less water-soluble toxins in both the starch and gluten fractions [357]. DON levels in the gluten fraction following wet milling of flour were only 15% of the level in the original wheat grain [387]. Less-water-soluble mycotoxins, however, can be concentrated in the gluten, with OTA a particular problem [338].

### 5.5. End-Product Utilization

The usual goal is to remove all mycotoxins before end-product processing and utilization. When that fails to happen, there are some actions that may reduce some of the contamination present. Some actions are inherent in the process, e.g., baking and extrusion, while others require addition of microbial cultures, enzymes or other products to have an effect. Some of these treatments may render the grain suitable solely for consumption by animals.

#### 5.5.1. Enzymes

Enzymes commonly are used in food processing, but their efficacy for mycotoxin degradation in a food context has not been extensively explored [388]. Neither have any enzyme treatments been approved specifically for decontaminating mycotoxins in foods. The range of effective enzymes is quite large [389]. Laccases and peroxidases can attack a wide range of mycotoxins, with most other enzymes being more specific in their activity. Specific studies have been performed of enzymes to degrade DON, ZEA and OTA. The number of such studies is usually few, the enzymes not always characterized in detail, and products of mycotoxin degradation usually characterized in even less detail. Yet these enzymes are potentially a safe and important way of degrading mycotoxins in the end products of the grain chain. Some enzymes were initially isolated as trans-genes that could be transformed into host plants and destroy mycotoxins almost as quickly as they were synthesized. In other cases, only the microorganism responsible for the degradation (see also Section 5.5.2 on Fermentation) has been evaluated.

Enzymatic degradation of DON is relatively poorly studied, even though the enzymes responsible for the process could be important additives for both baking and malting processes. The known DON-degrading enzymes are quite varied in their type and include glycosyltransferases, lipases and cytochrome P450 systems, among others [390]. In some cases, e.g., the glycosyltransferases, the detoxified end product may be converted back to DON under appropriate chemical conditions.

A variety of enzymes also are known to degrade ZEA [391]. One of these, a lactonohydrolase from *Clonostachys* (*Gliocladium*) *roseum* [392,393] breaks the lactone ring and inactivates the toxin. This enzyme has been patented [394] and a trans-gene encoding the enzyme expressed in several bacteria and fungi [391], including both *Saccharomyces* [395] and *Lactobacillus reuteri* [396], which is available commercially as a probiotic. More recently, a peroxiredoxin from an *Acinetobacter* has been isolated and the gene encoding it cloned and expressed in *E. coli* and *Pichia pastoris* [397]. This enzyme breaks ZEA into two pieces, neither of which has measurable estrogenic activity. As with DON, glycosyltransferases can transform ZEA to a zearalenone-glycoside (ZEA-G) that can easily be reconverted back to the original toxin under the proper environmental conditions. There are numerous microbial systems with cell-free extracts that can be used to degrade ZEA [391], but the enzymes, degradatory pathways and the breakdown products are not yet well characterized.

Several different classes of enzymes, including peptidases [388,398], lipases [399], and a patented amidase [400], can reduce the toxicity of OTA. The primary target is usually an amide bond that when broken yields phenylalanine and ochratoxin α, which is much less toxic than OTA. The number of studies of purified enzymes for degrading OTA is significantly less than the number of studies that identify a microbial strain(s) capable of degrading the toxin [401].

#### 5.5.2. Fermentation

Fermentation to remove mycotoxins can involve bacteria or fungi. The fermentation may occur in or as part of the final food product or in a reaction that proceeds the foods’ final preparation. Removal can be through absorption to a cell wall, sequestration within a cell, or degradation or biotransformation of the mycotoxin of interest into a different, less toxic, compound. Cells that have absorbed or sequestered toxins usually are removed before the final food is ready for consumption. In some cases, mixing a microbial strain with an identified enzyme can be synergistically efficacious in reducing the amount of toxin in the end food product.

Numerous fungi and bacteria can degrade DON both aerobically and anaerobically [390]). Not all of these organisms are suitable for incorporation into foods. Strains of the yeast *Geotrichum* can adsorb DON and remove it from grain used to brew beer [402], and strains of the yeasts *Geotrichum*, *Metschnikowsa* and *Rhodotorula* can adsorb the toxin and effectively detoxify animal feeds [403]. Amongst bacteria, several *Lactobacillus* and *Propionibacterium* strains can adsorb DON [404,405] with dead bacteria often as effective as living cells, suggesting that the toxin is sequestered by the cell wall. There are numerous bacterial strains and species that can degrade DON, and at least five potential degradation pathways have been identified [390]. In some cases, the bacteria used are from animal intestinal microbial communities and might potentially be useful in food and animal feed applications.

ZEA also can be sequestered by absorption to microorganisms, including *Saccharomyces* [406] and bacteria such as *Lactobacillus* and *Propionibacterium* [405,407]. As with DON, the binding is as effective with dead cells as it is living ones. In most cases, the efficiency of removal of ZEA is higher than the efficiency of the removal of DON. Cell wall composition, e.g., the amount of β-d-glucan present, may be particularly important for determining the binding efficiency of ZEA [408]. Microbial fermentation degradation of ZEA is not as well characterized as is degradation of DON. There are several well-characterized enzymes (Section 5.5.1.) that can be used with a microbial absorbent, often yeast cell walls, to effectively reduce ZEA exposure, but the living cell systems usually are not well-enough characterized for use in a food or feed setting [391].

OTA adsorption and degradation are well documented [401]. Adsorption by yeast cell walls is similar in nature to that described for both DON and ZEA. Cell walls of several strains of *Lactobacillus* also can bind OTA, but they are not as efficient in removing the toxin as are the yeast cell walls. Many microorganisms can degrade OTA, and some are used as whole cells in animal feed products. Microbial fermentations to remove OTA from human foods commonly focus on wine and grape juice.

##### Brewing

Small grains are an important component of the brewing process for beer. Most mycotoxins can survive the brewing process and end up in the final beer. DON can be found both as DON and at the same or higher levels as DON-3G [409,410]. Toxin levels decrease when the grain is steeped by as much as 10-fold before increasing beginning with germination to levels as much as five times higher than the initial level in the final malt [409]. These levels decrease through various steps of the brewing process and the levels in the final beer usually are similar to those seen in the unprocessed grain prior to malting. ZEA levels also increase when the grain is germinated in the malting process, with 80–90% of the ZEA in the malt grist continuing through to the final beer [411]. Ergot alkaloids usually are lost during the brewing process, with approximately 2% of the alkaloids present in malt samples contaminated with sclerotia being detected in the final beer [412]. OTA is not a common contaminant, and often is barely detectable, in beer [413].

#### 5.5.3. Bread and Other Baked Goods

An important use for small grains is in breads and other baked goods. The products produced are a complicated matrix that can bind some toxins and convert them to masked forms. Analytical recovery is thus an important variable, since recovery rates can vary from ~80% to ~110% [338], depending on the substrate, temperature, moisture and product history.

Temperature is an important variable in determining mycotoxin stability through the baking process. Reports of mycotoxin degradation during the baking process are numerous and the range of degradation observed quite high. For DON, the reductions may be as much as 60%, e.g., Abbas et al. [414], although values of 20–30%, e.g., Lancova et al. [415] and Neira et al. [416], are more common, while increases can be as much as 60–90%, e.g., Zhang and Wang [417]. Clearly, much work remains to be performed to identify the factors responsible for these extraordinarily different results. Increases could result from flour treatments or dough processing that free DON from cell wall or protein components of the flour or that break down DON-3G, releasing the DON in a detectable form. Baked products are not uniform across the product in terms of temperatures attained and for how long. For example, crusts usually experience much higher heats than do the interior crumb parts of the bread, and crackers and cookies are usually much thinner than breads, which reduces the variation in degradation across these products. Mycotoxins introduced as pure standards into the material may fare differently than those introduced as natural contaminants of the flour or other bread components. Length of exposure to heat was as important, or more important, than the highest temperature experienced by the product during the baking process [338,418]. Toasting bread more effectively reduces DON levels at lower oven temperatures than baking does [419].

Baking bread usually involves a fermentation step as well, most commonly with *Saccharomyces* and less commonly with *Lactobacillus* for sour dough. The fermentation steps usually acidify the mixture and can release mycotoxins from the flour or convert toxins from one form to another. As with baking, the reported effects of this step are quite variable, although there generally are more reports of increases in DON levels than decreases following this step. Reductions in the remaining processing steps often reduce the mycotoxin levels back to within a few percent of what it was in the original flour used for the process. Both pH and the suite of microorganisms in the fermentation mixture can affect the fate of any mycotoxins that might be present.

In addition to flour, other ingredients may be added to the dough before or during fermentation. These may include one or more of baking soda, ammonium salts (usually carbonate or phosphate), antioxidants, emulsifiers, malt powder, sodium bisulfite, sucrose, and whey. Enzymes such as α-amylase, cellulase, glucose oxidase, lipase, protease, and xylanase also may be added. Depending on conditions, the additive may have a significant effect on toxin detection and availability. Given the large number of additives and the great variation in technique, the data to date are primarily anecdotal in nature rather than systemic evaluations of a particular product, or class of products. Thus, the current status of knowledge regarding supplemental chemical leavening agents and bakery improvers is insufficient to determine a priori which additive, or combination of additives, has the greatest impact on the reduction in toxin levels.

#### 5.5.4. Pasta

When pasta is made, wheat flour usually is mixed with other ingredients to form a dough that is extruded through a noodle machine of some sort to give a final version of the raw pasta. None of these steps changes the mycotoxin levels of the ingredients used to make the original pasta dough. Thus, minimizing mycotoxin contamination of flour to be used to make pasta is very important in reducing human exposure to these contaminants.

Not all pasta/noodle production processes, however, result in no change in mycotoxin contamination levels. Production of instant noodles for both Western and Asian markets can reduce DON contamination by more than 50% [420] and DON-3G levels by as much as 87% [421]. Reducing DON-3G levels does not decrease the DON detected in the final product. Instant noodle production can also lower the amount of OTA present by up to 20% [386], and the amount of ZEA present by >60% [421].

Cooking dried pasta usually reduces toxin contamination because the water absorbed by the pasta during the cooking process increases the weight of the pasta. Thus, values for these studies must be corrected for moisture changes in the final product. Water-soluble toxins may elute into the cooking water as well. Reductions from elution may be as much as 65–75% of the DON present, depending upon the amount of water in which the product was cooked [422] and the length of time for which it was cooked [423]. Nearly 50% of the NIV present [424], virtually all of the enniatins [425], up to 50% of the moniliformin [426], and one third of the OTA [427] can be removed from pasta during the cooking process. The water in which pasta is cooked should be discarded and the pasta rinsed thoroughly to remove any toxins eluted into the cooking water that might have remained with the final product.

#### 5.5.5. Extrusion

Extrusion cooking is performed with wheat, but has been much more widely studied in maize. The literature on the effects of extrusion cooking on mycotoxins in wheat is not as abundant as it is for maize. Extrusion is important for processing wheat and other small grains into human foods and animal feeds. Flour and water are the basic ingredients with sugar and other additives included as needed. The process includes heating the mixture and then subjecting the heated mixture to shear and pressure resulting from a rotating screw with the final product usually pushed out through a die that molds the final product. After extrusion the final product is dried, often to less than 4% moisture content.

The combination of heat, pressure and shear can degrade mycotoxins, but the amount of degradation is heavily dependent upon the conditions used. In general, raising temperature and reducing screw speed to increase the exposure to the higher temperature increases degradation of thermo-labile mycotoxins. Loss of DON from wheat grits can be as much as 60% [428] and the loss of OTA from whole meal flour as much as 40% [429]. To the extent that these conditions can break glycosidic linkages, toxin levels may also increase as masked versions of toxins are exposed [430].

#### 5.5.6. Binders

Binders, such as montmorillonite clays are commonly added to animal feeds containing maize contaminated with aflatoxins. The binders reduce or prevent adsorption of the aflatoxin by the animal as the feed passes through the gut. Binders are not widely used with wheat-based animal feeds. One study with low-quality feeds based on wheat naturally contaminated with aflatoxin, OTA and ZEA included silicoglycidol as a binder and improved daily weight gain in swine [431]. A second study of a wheat/barley feed artificially contaminated with DON found that a commercially available binder composed of bentonite and clinoptilolite clays and yeast cell walls yielded similar results [432]. A yeast glucomannan binder did not alter DON uptake from naturally contaminated feed by turkey poults [433]. Biomin^®^ II when included with barley/alfalfa diet reduced the uptake of naturally occurring ergot alkaloids by lambs [434]. Graphene oxide is 90% efficient in the removal of ZEA in in vitro tests and should be considered for in vivo tests in the near future [435].

## 6. Nominal Group Discussions

The Nominal Group technique is a facilitated discussion process that supports equal input from participants [436]. The process generates numerous ideas and includes a mechanism for ranking them. From the list of ideas, particular ideas and general trends of note often can be identified. This technique has been used previously to identify priority areas for research on mycotoxins [36,437,438]. Six questions were selected for discussion (Table 1).

### 6.1. Strategy and Methodology

To help identify priorities for future research on mycotoxins and small grains, a Nominal Group discussion session was held on 18 September 2018 following the 2nd MycoKey conference in Wuhan, China. Roundtable participants were selected by MycoKey project leaders and divided into two groups: NG1—Emerson Del Ponte (Moderator), Florence Forget (Rapporteur), Antonio Moretti, Mark Sumarah, Zang Xu, Liu Yang, and Liao Yucai; and NG2—Sofia Chulze (Moderator), Hao Zhang (Rapporteur), Kris Audenart, Tom Gräfenhan, Chen Huaigu, Ákos Mesterházy, Pawan Singh, and Theo van der Lee. A moderator, who managed the discussion within the group, and a rapporteur who recorded ideas on a flip chart were identified for each group.

Both groups addressed the same set of questions (Table 1). The discussion process follows that outlined in Leslie et al. (36) and is composed of five basic steps:Silent generation of ideas.Sharing and recording of unique ideas.Idea explanation and potential modification.Voting and ranking. Each participant ranks the five most important answers for the question on the flip chart list, with the most important answer being given a “5”. The second choice answer receives a “4”, the next a “3”, and so on. Thus, for each response two numbers are generated—the number of individuals who considered the response amongst the five most important answers to the questions and a weighted ranking number that is the sum of the rankings of the participants who selected the response as one of the five most important.Presentation of results. Results from the discussions are presented on a question by question basis in Table 2, Table 3, Table 4 and Table 5. Results for pre-harvest questions 1–3 are summarized in a single table (Table 2), with results for the other questions each presented in individual tables (Table 3, Table 4 and Table 5). Within each table, responses that were selected by both groups are given first and followed by those selected by a single group. Responses within a set are ordered based on the number of individuals selecting the response, and then by the summed weight of the responses. Responses given in a group, but not included on any individual’s list of the top five (so no weighted score), are denoted with a “●” to enable distinction between an unweighted response and a response that was absent, which is left blank (“–”).

### 6.2. Nominal Group Questions 1–3

The first three questions all address pre-harvest toxin contamination problems and differ only in the toxins/fungi of interest (Table 2). Addressing these problems must be integrated, so responses to these questions are grouped in a single table to enable critical issues across toxins to be more readily identified. Climate-based differences are incorporated into these results per se and could cause rearrangement of priorities. There is much more known about the control of DON and ZEA (Q1) than there is about the other toxins, which provides a different weighting to answers than for the other two questions.

Across all three potential pre-harvest problems, 28 topics were identified in the groups, with all but three identified as a top five priority by at least one person for at least one of the three toxin problems (Table 2). In broad areas, the responses could be grouped as breeding, disease management, agronomy, research issues, fungal population size and composition, farmer support, and production system sustainability. Significant research issues (9, 10, 12–15, 22 and 25) were scattered across all of the other topics indicating that continuing work in all fields at both basic and applied levels is needed.

Breeding (1) was the most important of all responses, with the breeding programs focused on disease resistance. Use of GMO methods (22) to improve resistance was mentioned, but received relatively little support, probably representing the current regulatory climate within the EU. Research into plant physiology, morphology, and stress responses (10) could lead to new breeding targets. Disease management received significant support, especially for fungicide application (2) for all three systems. For the non-*Fusarium* toxins, fungicides may be the best hope of solving pre-harvest problems since the likelihood of establishing a breeding program that focuses on these toxins and diseases is unlikely until the losses due to these entities are better characterized or the toxins are regulated. Conducting tests for fungicide efficacy in parallel for multiple diseases should be possible without large expense and could bring benefits of the research performed on *Fusarium* to bear on other fungi of potential economic importance. Integrating fungicide management with appropriate host lines to obtain maximum disease resistance is a clear priority, and is most likely to occur with the *Fusarium* toxins.

Additionally related to fungicide application was research into mycotoxin biosynthesis inhibitors (9) and the development of new fungicides (14). Allies of fungicides—IPM (5) and biological control (6)—also were among the top 10 responses. All of these topics were brought up in both Nominal Groups as solutions for all three disease groups. These topics were not equally important for all three disease groups, however. For example, IPM was relatively more important for the “other” group, and the need for new fungicides and mycotoxin biosynthesis inhibitors were more important for the control of DON and ZEA.

The other significant disease management topic was disease forecasting (8) and the associated decision support systems (16), which, presumably, would require additional farmer training (17) to be effective. Disease forecasting results are particularly important in managing the timing of fungicide applications to reduce disease incidence and toxin production. New/improved diagnostics (20) were important for T-2 and HT-2 (Q2) and for the “other” group of toxins (Q3), but not for DON and ZEA (Q1).

Agronomy plays a major role in reducing disease and toxin production. Tillage (3), crop residue management (4) and crop rotation (7) were all recognized as important factors by both groups for all three toxin groups, but there remains much work to be performed to understand the nuances of the consequences of variation in the implementation of these strategies. Planting date (11, except for the “other” group) was also an important factor. Numerous additional factors received some level of support: planting site location (18—for Q2 and Q3), intercropping (19—for Q3), fertilizer level (21—for Q1), seed treatment (23—for Q3), and herbicide/weed control (24—for Q3). Planting density (26—Q3) was mentioned in one group (NG2), but was not a top five priority for anyone in that group.

Reducing fungal inoculum (27) is a goal of many agronomic practices. Studies involving fungi are all basic/applied research questions that lead to new breeding targets, changes in agronomic practices, or an understanding of fungal physiology (15) or more specifically toxin production conditions (13). Managing fungal populations, which can vary by location, requires an understanding of species distribution and population biology (12). The basic nature of this research often makes it a target for cutting in applied programs focused on toxin and disease reduction. Yet, this research is a critical component of an integrated successful program because of the insights the results obtained can offer to the development of novel approaches for the control of these problems.

In summary, there are multiple facets to pre-harvest toxin contamination control that must be considered in a toxin management program, and an integrated management program is essential. The components are inter-dependent and the degree of importance of individual components varies, but changing just one factor in isolation is unlikely to lead to major reductions in toxin contamination. In some cases, further research is needed to better define the problem and to identify areas to target for further research. As responses to these toxin contamination problems are implemented, a team approach is essential to assure that breeding, disease management, agronomic and fungal population/biology have all been considered and that the resulting solutions are both economically feasible (22) and sustainable (28).

### 6.3. Nominal Group Question 4

Harvest and post-harvest are crucial times since post-harvest storage may be longer than the time the grain was in the field. Toxin contamination begins in the field and may increase in storage if the grain is not properly treated and maintained. New pests, pathogens and toxins also may be introduced. Problems that occur early in the post-harvest process may magnify existing problems or enable new ones later in the process. Post-harvest protocols generally are well established, and for developed countries variations on the process need to be considered, while in less-developed countries the process is not as well defined, and its efficacy may be limited by the energy available for processes such as drying and cooling. The discussion below focuses on developed countries. There were 23 responses to this question, of which 20 were among the five most important for at least one member of a group, and nine were brought up for discussion in both groups (Table 3).

Harvest is the first step in the process, although it was not viewed as the most important one. Adopting suitable technology (16) is important and adjusting combine settings (10) to blow out tombstone kernels with the chaff might be the most important. Timing of harvest (20) also is important, and toxin testing and forecasting (18) implemented as part of a decision support system (14) can help determine the optimum time for harvest.

Transport from the field to storage is the next step, but also was not highly rated. Climate control (9) during transport was the only topic in this area brought up in both groups, and hygiene (13), which was brought up in only a single group, was the only one receiving top five votes. The speed of transport (22) was brought up in one group but was not on any top five list.

Prior to storage, multiple events can help reduce toxin contamination or its potential while in storage. The most important event is to dry the grain to a moisture level < 12%, but drying as a process is named here only through the need for drying equipment (15). Seed cleaning (11) to remove foreign debris and sorting, either mechanical (3) or based on toxin detected (5), also occurs at this time and can affect both storage conditions and the length of time that the harvested grain can be stored. Toxin decontamination (6), e.g., ozonation, and disinfestation (11) or biological control treatments (7) usually occur prior to storage as well. Four of the top seven rated events occur at this pre-storage stage and suggest that continuing research to optimize these procedures is important.

Storage conditions had two separate sets of concerns. The first set was for the specific conditions themselves, which were the other three of the top seven ranked events—reduced humidity (1), climate control (2), and insect control (4). These well-known issues were clearly on top of the participants’ minds. Other topics included lowering storage temperatures (19), which could be part of either of the previously listed reduced humidity (1) and climate control (2) topics, separating grains of different quality during storage (17), and limiting the time the grain spends in storage (23). The second set of concerns was on approaches to storage technology and monitoring of the grain while in storage. New technology (8) was needed and could involve HAACP protocols (12) or new decision support systems (14) for both on-farm and commercial storage facilities. Better sensors for monitoring stored grain should also increase our understanding of the basic physiology of the seeds being stored.

An effective means of reducing mycotoxin contamination is blending (21), wherein grain with higher levels of toxin is mixed with grain with lower levels of contamination, with the usual goal being for the entire blended lot to have a contamination level below a particular cutoff. This method was suggested in one group, but was not on any group member’s top five list. The practice is illegal in most jurisdictions and the question should probably be how to detect when such a process has occurred rather than to encourage its use to reduce contamination levels.

### 6.4. Nominal Group Question 5

Processing and decontamination/detoxification (Table 4) are intimately intertwined with the storage issues previously addressed above in Question 4 (Table 3). Indeed, some responses are common to both tables, e.g., sorting, blending, and seed cleaning. The types of things that could be performed during processing or to decontaminate/detoxify (Table 4) can be placed into three broad categories—rearranging all or parts of the grain (1, 8, 10 and 13), managing information about the grain (9, 11, 12, 14, and 16), and things that can be done to the grain (2–7, 15 and 17) to change it in some manner. The potential rearrangements are usually based on sorting (8). Sorting may remove contaminants through seed cleaning (10) or by rejecting grain contaminated above a specified threshold. Blending (8) falls into this category as well, but is useful only when it can be performed legally.

There are many ways to monitor grain during storage or processing that can help reduce contamination. Monitoring toxin contamination before (9), and during processing (17) can provide opportunities to alter control process parameters (11) that ultimately reduce contamination. Having a regulatory target (14) incentivizes the minimization of toxins present. Training farmers (12) and those working at storage silos and processing plants to implement best practices to reduce contamination levels also is important. The most important mycotoxins are stable through many physical treatments, e.g., baking, boiling (5), and irradiation (17), although some can be degraded when material is extruded (15). All of these processes, however, can kill viable fungi present in or on the seed, and effectively decontaminate the material.

As toxins are not uniformly distributed within a seed, processes such as milling and dehulling (2), which remove outer layers, or washing (4), which may remove parts of fungal colonies and pieces of seed to which they are attached, can be effective as well. Processes that target toxins, e.g., enzymatic degradation (3) or fermentation (6), are less well studied and may render the grain unfit for some purposes. Mixing grain with toxin binders prevents the toxins from being adsorbed in the gut after the contaminated material has been consumed, but does not reduce the amount of toxin present in the grain. Chemical processes, e.g., chlorination or ozonation, neither of which was brought up as a decontamination technique, also are effective, but the permitted uses for grain treated with these techniques may be limited and of relatively low economic value. Identifying effective decontamination techniques and methods, and uses for potentially heavily contaminated grain remains an important area of great commercial interest

### 6.5. Nominal Group Question 6

Responses to this Nominal Group question (Table 5) were the most numerous and diverse. Responses spanned the gamut of areas for research and applications to manage mycotoxins in wheat. These responses show the complexity of the problem and the interdependence of different approaches to its amelioration. Thus, work needs to continue on everything from basic research on fungal populations to improved host lines and apps that can be used to monitor and manage disease and toxin problems both in the field and in storage.

Responses provided can be clustered, but there was no single theme that dominated the discussion. Only 6/33 responses were common to both groups and all but 9 ideas were included as a top five priority for at least one participant. To help organize and focus efforts there were suggestions that underlying questions needed better definition (9, 19), that available background information needed to be more readily accessible (24, 30), and that relevant information from closely related areas (25, 26) needed to be considered as part of the decision-making process.

Management tools were the most frequently identified general group of responses and amongst the most highly rated. Some management tools were aimed at farmers, with an app (1) and a decision support system (5) both identified as important in both groups. The provision of additional training (13) and development of best management practices (14) also received support. The ability to assess risk (12) is an important component of any management strategy, with disease/toxin forecasting systems (4) and risk maps for toxin contamination (10) both playing potentially important roles in the risk assessment and management process. Management of post-harvest storage issues, e.g., warehouses (15) and storage silos (22), received priority votes but were not highly ranked.

In the field (pre-harvest), two general areas were emphasized. The first was development and availability of appropriate host lines (2, 7, 33) in which the physiological basis of resistance was understood (28, 29), and phenotypes could be rapidly identified (11). The second area was disease control. Forecasting models (4) and decision support systems (1, 5) both help identify optimum disease control strategies. Work is needed to develop new chemical, i.e., fungicide (3, 6), controls that work with biological controls (18) and as part of an IPM program. Disease control generally occurs while the crop is in the field, but inoculum control/reduction also can occur when the field is fallow (16). Host resistance and chemical/biological control require current information on the fungal species present (8) and an understanding of the interactions between toxigenic and beneficial fungi (17). The need for new agronomic information was noted (20), but was not a high priority.

Post-harvest issues included some distinct issues and some overlap with pre-harvest issues (3, 4, 5, 10, 11, 12, 14, 17, 31), although the context differs. In terms of management, risk assessment tools (4, 10, 12) are important for direct storage and potential end uses, as well as management of warehouses (15) and silos (22). Detection of regulated toxins (11) is critical, as is the ability to detect new and emerging toxins (21). Methods to decontaminate grain (3, 20) will be important, as will modified regulations that allow safely decontaminated grain to be used in human foods and not just for animal feeds. Drying (32), especially in warm, humid environments, is an important technology that needs to be as energy efficient as possible. Sorting grain (30) depends on the ability to rapidly phenotype it (11), especially for mycotoxin contamination (21), and to have clear standards for different qualities of grain and a sufficient price differential to justify the cost of segregating, tracking, marketing, and managing the grain in the different classes.

### 6.6. Nominal Group Questions Discussion

The results of the Nominal Group discussions have a message that is more nuanced than what is readily discerned from the available literature. Control measures and enablers of mycotoxin production do not act independently, but instead there are multiple, often very complicated interactions. Experimentally, it often is difficult to test more than two or three variables at the same time, and the complexity of the interactions occurring may not be clear. However, the Nominal Group opinions reflect the complexity of the interactions, and the responses indicate that host resistance is critical, but that other factors such as the previous crop planted, tillage, fungicide treatments, storage parameters, and many other measures also have important roles in determining final grain quality. This network of interactions is important, since complete host resistance to toxin is not available. Additionally, analytical methods that provide rapid, reliable results and can detect multiple mycotoxins are fundamental as these methods are used at every link in the small-grain chain from breeding through growth in the field to post-harvest. Without such methods, responses to potential contamination issues will be slow, erratic and not as effective as they could be if better information on the contamination that is occurring were more quickly available. The take-home message is that the small-grain chain needs to be treated as an interconnected system and that consequences of changes made in one part of the system may have important implications for other parts of the system and should not be made autonomously or in isolation.

## Figures and Tables

**Table 1 toxins-13-00725-t001:** Questions for Nominal Group discussion sessions.

No.	Question
	Identify effective measures for minimizing pre-harvest contamination of small grains by:
1	DON and ZEA
2	T-2 and HT-2 toxins
3	Other toxins, e.g., *Alternaria* toxins, ergot alkaloids and aflatoxins
4	Identify effective measures for minimizing mycotoxin contamination in small grains post-harvest
5	Identify processing steps and/or decontamination/detoxification actions to reduce mycotoxin content in small-grain products
6	Identify information to be generated or questions to be answered to help those involved in the small-grain chain continue to make progress in reducing mycotoxin contamination after the MycoKey project ends in 2020

**Table 2 toxins-13-00725-t002:** Answers to Nominal Group questions 1–3. Identify effective measures for minimizing pre-harvest contamination of small grains by: (Q1) DON and ZEA, (Q2) T-2 and HT-2 toxins, and (Q3) other toxins, e.g., *Alternaria* toxins, ergot alkaloids and aflatoxins.

Response Number	Q1				Q2				Q3				Response
	NG1		NG2		NG1		NG2		NG1		NG2		
	#¹	*S* ^2^	#	*S*	#	*S*	#	*S*	#	*S*	#	*S*	
1	7	25	10	37	3	15	8	34	3	10	3	8	Breeding for resistance
2	6	24	4	18	5	17	2	4	4	13	6	21	Fungicide application—timing and technology
3	3	9	1	2	3	9	3	7	3	10	2	6	Tillage
4	1	3	1	3	2	7	3	7	3	10	2	6	Crop residue management
5	●^3^	–^4^	2	7	●	–	3	7	2	3	2	5	IPM
6	3	3	1	2	1	2	1	4	3	7	●	–	Biological control
7	4	15	5	10	4	12	3	7	3	8	–	–	Crop rotation
8	4	9	5	17	2	4	5	8	–	–	–	–	Disease forecasting
9	2	4	3	5	2	4	–	–	2	6	–	–	Research: Mycotoxin biosynthesis inhibitors
10	●	–	1	1	–	–	1	4	–	–	4	13	Research: Plant physiology, morphology and stress
11	1	1	●	–	1	3	3	7	–	–	–	–	Diversify planting dates
12	–	–	●	–	–	–	8	25	–	–	6	17	Research: Fungal species distribution and population biology
13	–	–	–	–	–	–	2	5	3	14	5	12	Research: Toxin production conditions
14	2	4	3	5	–	–	5	15	–	–	–	–	Research: New fungicides
15	●	–	1	1	–	–	–	–	6	20	–	–	Research: Fungal physiology
16	●	–	●	–	–	–	–	–	–	–	5	17	Decision support systems
17	3	4	–	–	1	1	–	–	1	5	–	–	Additional farmer training
18	–	–	–	–	1	2	3	7	●	–	–	–	Planting site location
19	●	–	–	–	●	–	–	–	1	1	–	–	Intercropping
20	–	–	–	–	4	10	–	–	3	6	–	–	New/improved diagnostics
21	1	2	●	–	–	–	–	–	–	–	–	–	Alter/increase fertilizer
22	–	–	–	–	–	–	–	–	–	–	4	11	Research: Economic importance
23	–	–	–	–	–	–	–	–	–	–	3	10	Seed treatment
24	–	–	–	–	–	–	–	–	2	7	–	–	Herbicides/weed control
25	–	–	1	2	–	–	–	–	–	–	–	–	Research: GMO for resistance
26	–	–	–	–	–	–	–	–	–	–	●	–	Planting density
27	–	–	●	–	–	–	–	–	–	–	–	–	Reduce fungal inoculum
28	–	–	●	–	–	–	–	–	–	–	–	–	Production system sustainability

^1^ #—Number of participants ranking this response as one of the five most important. ^2^
*S*—Weighted priority score, with each voting member ranking their top five topics. Five points assigned to the most important response and one point to the least significant of the important responses. ^3^ ●—This response was provided by one or more members of the group when ideas were listed, but was not identified as one of the five most important responses by any member of the group. ^4^ “-–—This response was not provided by any member of the group.

**Table 3 toxins-13-00725-t003:** Responses to Nominal Group question no. 4: Identify measures that are effective for minimizing mycotoxin contamination in small grains during harvest and post-harvest.

Response Number	NG1		NG2		Response
	#¹	*S* ^2^	#	*S*	
1	5	20	5	14	Reduced humidity/water activity
2	2	8	7	27	Climate-controlled storage
3	3	11	3	12	Mechanical and physical seed sorting
4	2	3	3	6	Insect control
5	1	4	3	12	Seed sorting for toxin based on NIR
6	2	5	1	1	Toxin decontamination/degradation
7	2	5	1	1	Biological control
8	•	-	2	7	New storage technology
9	•	-	•	-	Climate control during transport
10	-	-	5	16	Combine settings
11	-	-	5	13	Seed cleaning/disinfestation
12	-	-	3	10	HAACP
13	3	7	-	-	Hygiene control during transport
14	-	-	3	5	Decision support system
15	-	-	2	5	Drying equipment
16	2	5	-	-	Harvesting technology
17	-	-	1	5	Separation during storage
18	1	3	-	-	Mycotoxin testing and forecasting
19	1	2	-	-	Lower temperature storage
20	1	1	-	-	Timing of harvest
21	•	-	-	-	Blending
22	•	-	-	-	Rapid transport from field to storage
23	•	-	-	-	Storage time limit

^1^ #—Number of participants ranking this response as one of the five most important. ^2^
*S*—Weighted priority score, with each voting member ranking their top five topics. Five points assigned to the most important response and one point to the least significant of the important responses. ^3^ ●—This response was provided by one or more members of the group when ideas were listed, but was not identified as one of the five most important responses by any member of the group. ^4^ “-”—This response was not provided by any member of the group.

**Table 4 toxins-13-00725-t004:** Responses to Nominal group question no. 5: Identify processing steps and/or decontamination/detoxification actions that could alter mycotoxin content in small-grain products.

Response Number	NG1		NG2		Response
	#¹	*S* ^2^	#	*S*	
1	4	16	8	33	Sorting
2	4	13	6	17	Milling and dehulling
3	5	12	4	11	Enzymatic decontamination (for feed)
4	4	12	3	6	Washing
5	3	8	3	10	Baking/Boiling/Heating
6	2	5	2	4	Bacterial or yeast fermentation
7	2	4	2	5	Mix with binders
8	•^3^	-^4^	3	7	Blending
9	-	-	3	12	Check toxin contamination prior to processing
10	2	8	-	-	Glume separation/seed cleaning
11	-	-	2	4	Control process parameters
12	2	4	-	-	Farmer/stakeholder training/education
13	1	5	-	-	Discarding
14	1	3	-	-	Establish regulatory process
15	-	-	•	-	Extrusion
16	•	-	-	-	Trace toxins through the process
17	•	-	-	-	Irradiation

^1^ #—Number of participants ranking this response as one of the five most important. ^2^
*S*—Weighted priority score, with each voting member ranking their top five topics. Five points assigned to the most important response and one point to the least significant of the important responses. ^3^ ●—This response was provided by one or more members of the group when ideas were listed, but was not identified as one of the five most important responses by any member of the group. ^4^ “-”—This response was not provided by any member of the group.

**Table 5 toxins-13-00725-t005:** Response to Nominal Group question no. 6: Identify information to be generated or questions to be answered now to help those involved in the small-grain chain continue to make progress in reducing mycotoxin contamination after the MycoKey project ends.

Response Number	NG1		NG2		Response
	#¹	*S* ^2^	#	*S*	
1	2	5	5	23	App for farmer/stakeholder use—knowledge translation and transfer kit
2	5	18	1	2	New resistant lines with pedigrees and catalog of current materials
3	4	12	1	2	Develop new products (preferably green) and supporting information
4	2	6	3	8	Forecasting system adaptable to climate change
5	5	17	•	-	Operation guidelines and decision support systems
6	•	-	4	11	Fungicide recommendations
7	-	-	4	14	New host resistance sources
8	4	12	-	-	Monitoring the global population of toxigenic fungi
9	-	-	3	8	What … why … when … how mycotoxins
10	-	-	2	8	Risk maps for toxin contamination
11	-	-	2	7	Rapid phenotyping technology
12	-	-	2	6	Risk assessment
13	-	-	2	2	Develop farmer/stakeholder training program
14	1	5	-	-	Information on best management practices
15	-	-	1	5	Smart warehouse management
16	1	3	-	-	Establish efficacy of biocontrol on residue inoculum
17	-	-	1	3	Characterize interaction between endogenous and toxigenic fungal populations
18	-	-	1	2	How to combine biological controls and chemical controls
19	-	-	1	2	List of “What if …?” questions
20	1	1	-	-	Chemicals for decontaminating grain
21	1	1	-	-	Efficient and effective detection of emerging toxins
22	1	1	-	-	Monitoring tools for silos
23	-	-	1	1	Better agronomic practices
24	-	-	1	1	Summarize historic field data
25	•	-	-	-	Information on future food production needs
26	•	-	-	-	Information on toxicology of other fungal secondary metabolites
27	•	-	-	-	New sorting methods
28	•	-	-	-	Resistance to emerging toxins
29	•	-	-	-	Understanding the relationship between plant physiology and resistance
30	-	-	•	-	Catalog of information available on the web
31	-	-	•	-	Improve IPM
32	-	-	•	-	Low-cost, energy-efficient drying technology
33	-	-	•	-	Multiple disease-resistant crops

^1^ #—Number of participants ranking this response as one of the five most important. ^2^
*S*—Weighted priority score, with each voting member ranking their top five topics. Five points assigned to the most important response and one point to the least significant of the important responses. ^3^ ●—This response was provided by one or more members of the group when ideas were listed, but was not identified as one of the five most important responses by any member of the group. ^4^ “-”—This response was not provided by any member of the group.

## Data Availability

All data collected from Nominal Group discussions are included in Table 2, Table 3, Table 4 and Table 5.

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
