# Peer review of "Key Global Actions for Mycotoxin Management in Wheat and Other Small Grains"

_toxins, 2021, doi:10.3390/toxins13100725_

Round 1
Reviewer 1 Report
I believe that this manuscript - review is a good contribution to this area. The manuscript provides the overview of currently pre- and post-harvest control strategies and Nominal Group discussion technique.
To further enhance the article, below suggestions are made.
- Line 269, please rewrite the sentence “Trichothecenes are a large diverse class of toxins produced by plants and fungi” as “Trichothecenes are a large diverse class of toxins produced by multiple genera of fungi including plant and insect pathogens”
- Lines 507-509, “This section…drawn” should be removed
- Line 757-758, “This section… in this manuscript” should be removed
- Lines 895-897, “This section…drawn” should be removed
- Line 907-908, “This section… in this manuscript” should be removed
- Line 1139-1140, “This section… in this manuscript” should be removed
- Line 1196, it is Table 2
Author Response
We thank the reviewer for the positive review and catching some minor issues that need to be corrected. The reviewer had some suggestions for revisions that are addressed below. The reviewer’s original comments are in black and our response follows in blue.
- Line 269, please rewrite the sentence “Trichothecenes are a large diverse class of toxins produced by plants and fungi” as “Trichothecenes are a large diverse class of toxins produced by multiple genera of fungi including plant and insect pathogens”
Rewritten.
- Lines 507-509, “This section…drawn” should be removed
The first two sentences of the paragraph have been deleted.
- Line 757-758, “This section… in this manuscript” should be removed
The first sentence of the paragraph has been deleted.
- Lines 895-897, “This section…drawn” should be removed
The first two sentences of the paragraph have been deleted.
- Line 907-908, “This section… in this manuscript” should be removed
The first sentence of the paragraph has been deleted.
- Line 1139-1140, “This section… in this manuscript” should be removed
The first sentence of the paragraph has been deleted.
- Line 1196, it is Table 2
Table number corrected.
Reviewer 2 Report
Dear Authors,
I found the manuscript as highly interesting and of high scientific soundness. The language is understandable and the review will be useful for the scientists and for practitioners dealing with the problem of mycotoxin contamination of grain. I would like to underline the great knowledge and experience of the Authors in this field and the enormous number of references. I think that the review will be a helpful comprehensive source of information and will gain a lot of citations.
I have still some suggestions:
- Authros write genera names in two ways - normally and in italics, please use only one style;
- The section in chapter 6 is not clear for the reader, I recommend to remove all tables and write a summary basing on the discussions' conclusions;
- You should read carefully the manuscript and remove some unnecessary fragments e.g. lines 1139-1140 (not pasted from the template).
With regards
Author Response
Reviewer no. 2
We are very happy that the reviewer found the manuscript to be very interesting and scientifically sound. The reviewer had some suggestions for revisions that are addressed below. We numbered these suggestions to make it easier to refer back to them in any future discussions. The reviewer’s original comments are in black and our response follows in blue.
- Authros write genera names in two ways - normally and in italics, please use only one style.
Journal style usually determines with names of genera are presented in Roman or Italics fonts. We tried to italicize the genera names everywhere they are used as a name for a fungal group, e.g., species binomials and when genus names are used in a similar manner. Many journals do not italicize a generic name when it is part of a common name or a disease name, e.g., the “Fusarium” in “Fusarium Head Blight” is part of a disease name and is in Roman type rather than italics. Also some headers are set in italics. In these instances if a genus name is used it is set in Roman type rather than italics. We have gone through the manuscript again looking for this kind of problem, and tried to fix them. If there are specific instances of concern we would be happy to fix them, as appropriate.
- The section in chapter 6 is not clear for the reader, I recommend to remove all tables and write a summary basing on the discussions' conclusions.
We think that removing the tables would be a major mistake. The first table (1) lists the questions discussed in a single easy-to find location. The remaining five tables provide all of the data upon which the discussion and conclusions in Section 6 are based. The summary requested by the reviewers is effectively sections 6.2-6.5. In each of these sections, the responses are grouped and their significance discussed in light of the questions posed. Similarities/differences between the two nominal groups are important for determining the significance of the responses. This information is lost if the tables are deleted. Similarly, similarities/differences between the different pathogen systems are most easily visualized by comparing the responses and their significance in Table 2. In conclusion, we think the discussion the reviewer would like is already present and that the tables are essential to be able to easily understand the discussion in sections 6.2-6.5. Thus, we have not made any changes to section 6 in response to this comment.
- You should read carefully the manuscript and remove some unnecessary fragments e.g. lines 1139-1140 (not pasted from the template).
The first sentence of the paragraph has been deleted, as well as additional problematic sentences of this sort identified by reviewer no. 1.